

# Global Assessment of Socio-Economic Impacts of Subnational Droughts: A Comparative Analysis of Combined Versus Single Drought Indicators

5  Sneha Kulkarni[1], Yohei Sawada[1], Yared Bayissa[2], Brian Wardlow[3]

1 Department of Civil Engineering, School of Engineering, The University of Tokyo, Tokyo 113-8656, Japan

2 Department of Ecology and Conservation Biology, Texas A&M University, College Station, TX 77843, USA

School of Natural Resources, Center for Advanced Land Management Information Technologies, University of Nebraska–Lincoln, Lincoln, NE 68583, USA

*Correspondence to*: snehakulkarni@g.ecc.u-tokyo.ac.jp

**Abstract.** The accurate assessment of the propagation of drought hazards to socio-economic impacts poses a significant challenge and is still less explored. To address this, we analyzed a sub-national disaster dataset called the Geocoded Disaster (GDIS) and evaluated the skills of multiple drought indices to pinpoint drought areas identified by GDIS. For the comparative analysis, a widely used Standardized Precipitation Index (SPI), Normalized Difference Vegetation Index (NDVI), Standardized

Soil Moisture Index (SSI), and Standardized Temperature Index (STI) were globally computed at the subnational scale. In addition, we developed a novel Combined Drought Indicator (CDI), which was generated by a weighted average of meteorological and agricultural anomalies. Out of 2142 drought events in 2001-2021 recorded by GDIS, NDVI, SSI, SPI, and STI identified 1867, 1770, 1740, and 1680 drought events, respectively. In terms of the skill to cover GDIS-documented drought events, CDI outperformed the other single-input-based drought indices and identified 1885 events. This emphasizes

the importance of using CDI to evaluate socio-economic drought risks and prioritize areas of greater concern.



# 1 Introduction

Droughts are a complex phenomenon and have profound, long-lasting impacts on various sectors, including agriculture, water resources, industry, energy, and socio-economic conditions. For instance, in August 2021, a severe drought event affected 52%
of crop yields in the Western United States, and in June 2019, Chennai, India, declared "day zero" due to almost no water remaining in all main reservoirs (Hossain Anika Tasnimand MacMurchy, 2022). Climate projections indicate a future trend of increased frequency and severity in drought occurrences worldwide, amplifying their impacts on various sectors (Vicente-Serrano et al., 2022).These examples emphasize the urgent need for enhanced drought preparedness mechanisms and accurate drought impact assessment techniques in the coming years. When addressing drought assessment, one of the major challenges
is in understanding the exact propagation of drought hazard (meteorological/agricultural or hydrological) to its socio-economic repercussions.

Previous studies often used proxies such as crop yield, reservoir storage, or vegetation dynamics to assess the societal impacts of droughts. Leng and Hall (2019) used crop yield sensitivity as a proxy for socio-economic impact and evaluated droughts in
major drought-prone countries. Similarly, Sawada et al. (2020), Bayissa et al. (2019), Udmale et al. (2020), Wu and Wilhite (2004) used agricultural productivity as a proxy to measure the socio-economic repercussions of droughts on a regional scale in North Africa, Ethiopia, India, and the USA, respectively. The water availability in reservoirs during drought periods can provide insights into on-ground socio-economic stress conditions; hence,Tiwari and Mishra (2019) Wu et al. (2018) considered reservoir storage as a proxy for socio-economic impacts and identified drought-affected areas in India and China. However,
the direct link between drought hazards and their socio-economic repercussions is still underexplored. One major reason for this gap is the lack of available socio-economic impact data. Until a few years ago, the U.S. Drought Impact Reporter, the European Drought Impact Report Inventory, or newspaper information were the only direct socio-economic impact assessment methods available. However, these tools were region-specific and did not offer global coverage. To understand global disaster vulnerability and risk, many previous works, such as  Below et al. (2007), Panwar and Sen (2020) Shen and Hwang (2019),
used the Emergency Event Datasets (EM-DAT: Guha-Sapir D. et al. 2014). However, the use of this dataset in text format presents limitations, highlighting the need for a more comprehensive, globally inclusive dataset to better understand the socio-economic impact of droughts.

Recently, the Geocoded Disaster (GDIS) dataset has been developed based on EM-DAT, offering geocoded disaster locations
at a subnational level (Rosvold and H. Buhaug., 2021). By addressing the limitations of EM-DAT, the GDIS dataset provides detailed information on socio-economically affected areas and administrative units in GIS polygon format. In this paper, we used this newly developed GDIS dataset, and show that it enables us to explore the less understood link between drought hazards and their socio-economic repercussions more accurately and comprehensively.



Droughts do not have a specific universally accepted definition. They are defined based on their association with different sectors and unique characteristics. Based on drought types and sectors, a number of drought assessment techniques and indices have been used worldwide to quantify and monitor drought conditions. To assess meteorological droughts, several indices such as the Standardized Precipitation Index (SPI)(Mckee et al., 1993), Standardized Precipitation Evaporation Index (SPEI)(Vicente-Serrano et al., 2010), and Temperature Condition Index (TCI)(Kogan, 1995) were extensively used. The World

Meteorological Organization (WMO) recommends (Svoboda et al., 2012) SPI, which is widely used in many regions. For example, Bhunia et al. (2020), Blain et al. (2022), Kazemzadeh et al. (2022), Livada and Assimakopoulos, (2007), Zhang et al. (2012), used SPI to assess drought over Greece, United Kingdom, Iran, India, and China, respectively. SPEI has been used in drought projection studies across the United Kingdom (Reyniers et al., 2023), the United States(Costanza et al., 2023), and the global (Vicente-Serrano et al., 2022) level, indicating an increase in severe drought events in these areas in upcoming years.

To assess agricultural droughts, the Standardized Soil Moisture Index (SMI), Vegetation Condition Index (VCI), Normalized Difference Vegetation Index (NDVI), and Vegetation Health Index (VHI) have been widely recognized as effective indicators. Ding et al. (2022), Sandeep et al. (2021), Tao et al. (2021) used the NDVI-based drought indices to perform detailed assessments of vegetation health and vigor in Australia, India, and China, respectively. Grillaki (2019) applied SMI to reveal a significant rise in drought severity across Europe in recent years. Liu et al. (2022) investigated soil moisture dynamics in

East Africa using a space-time perspective to identify the underlying causes of drought. By a novel Eco-hydrological reanalysis, Sawada (2018) quantified the drought propagation from soil moisture to vegetation across the globe, highlighting that deeper soil layers exhibit delayed recovery from stressful conditions compared to shallow layers. To assess hydrological droughts and their impacts on water resources and ecosystems, commonly used metrics include the Standardized Streamflow Index (SSI), Reservoir Storage Index (RSI), and Standardized Runoff Index (SRI). To understand surface water security,

Mishra (2020) examined long-term trends in hydrological droughts across India using SRI. Forootan et al. (2019) conducted a global assessment of hydrological droughts, highlighting the strong regional impact of the North Atlantic Oscillation and Indian Ocean Dipole.

The studies mentioned above provided drought indicators that were based primarily on a single input variable or were sector-

specific (either meteorological, agricultural, or hydrological). However, some works suggested that drought is a much more complex and multiphase phenomenon, resulting from various factors rather than just a single parameter(Jiao et al., 2019a; Kulkarni and Gedam, 2018; Sepulcre-Canto et al., 2012). Hence, to understand droughts more accurately, the integration of multiple variables is needed, and it has a higher potential than single-variable-based traditional drought indices. This integrated approach can provide a more comprehensive understanding of droughts, considering the relationships between various

contributing factors and the resulting impacts across different sectors. Recognizing the significance of combining multiple variables in drought monitoring, Svoboda et al. (2002) developed a drought monitoring system (US Drought Monitor) for the United States, which has been extensively used for regular practices in the United States. More recently, the near real-time





Vegetation Drought Response Index (VegDRI) was developed and implemented in South Korea (Nam et al., 2018) and the USA (Brown et al., 2008), demonstrating more detailed and improved spatial drought patterns compared to multi-variable

based drought indicators. This VegDRI was developed by integrating eight climatic and biophysical datasets (SPI, Palmer drought severity index, performance of average seasonal greenness, start of seasonal anomalies, soil availability water capacity, irrigated agriculture, and ecological regions). In 2023, Guillory et al. (2023) developed the Australian Drought Monitor, integrating SPI, NDVI, soil moisture, and evapotranspiration, which has become a valuable tool in Queensland's official drought declaration process. Bayissa et al. (2019b), Huang et al. (2019), Kulkarni et al. (2020a), and Sepulcre-Canto et al.

(2012) have developed and tested Combine Drought Indicators (CDI), demonstrating higher accuracy over Ethiopia, India, China, and Europe, respectively.

Despite many works on single-variable and multi-variable drought indicators, very few have investigated how useful these indices are in globally exploring the links between drought hazards and their socio-economic repercussions. In this study, we

globally applied four commonly used drought indicators—SPI, STI, NDVI, and SSI and compared these four traditional indices with GDIS to evaluate how well they represent the socio-economic impacts of droughts. Then, inspired by the successful regional examples of CDI, we also developed a new CDI on a global scale using two meteorological (rainfall and temperature) and two agricultural (soil moisture and NDVI) variables. To the best of our knowledge, very few studies (Hao et al., 2014; Sánchez et al., 2018; Wang and Sun, 2023) have considered a global scale for their combined drought indicators,

and none have compared and demonstrated the superior capability of such indicators over single-parameter-based traditional indices in global drought assessment. In addition, no studies have assessed the skill of drought indicators in identifying sub-national socio-economic impacts globally, although the sub-national disaster dataset (i.e., GDIS) recently made it possible. In the present paper, we addressed the following key points.

1. Understanding the link between global drought hazards and their socio-economic impacts at the subnational scale
using GDIS data.

2. Developing a new global combine drought indicator to enhance the precision and reliability of drought assessment (agro-climatological as well as socio-economic) and assessing its performance in detecting GDIS drought events.

3. Checking the performance of the commonly used traditional drought indicators (SPI, STI, NDVI, and SSI) in identifying sub-national socio-economic impacts of droughts (i.e., their association with GDIS).


To achieve the stated objectives, our study was structured into three main components. Initially, we conducted an analysis of global GDIS data to identify and select drought events at the subnational level. Later, a new global CDI was developed, and its performance in identifying GDIS events was assessed. Following this, we evaluated the effectiveness of four single-variable-based traditional indicators (SPI, SSI, STI, and NDVI) in detecting the GDIS events compared to CDI. As a result,

our research produced a comprehensive global framework for assessing drought impact, integrating agroclimatology hazard





data with socio-economic impacts. This framework offers potential benefits for drought-prone regions worldwide, facilitating improved drought management strategies and informed policy and decision-making processes.

## 2 Data

We used global observation and simulation data of rainfall, temperature, soil moisture, and NDVI to provide various drought indicators. These data sets were sourced from a combination of satellites and models and cover the time span from 2001 to 2021 on a monthly temporal scale. All the datasets had varying spatial resolutions. To address this disparity, we applied an Inverse Distance Weighted (IDW) method and rescaled all the datasets to a consistent resolution of 0.10°*0.10°. All datasets were selected based on factors such as long-term data availability, accessibility, reliability, high spatial resolution, and overall data accessibility. To explore the linkage between drought indicators and socio-economic impacts, we used subnational disaster data. The subsequent paragraphs provide a detailed description of each dataset used in the study, and Table 1 outlines the respective sources of acquisition.

### 2.1 Rainfall

The monthly rainfall data were obtained from the Climate Hazards Group Infrared Precipitation with Station data (CHIRPS), developed by the U.S. Geological Survey(CHIRPS: Rainfall Estimates from Rain Gauge and Satellite Observations | Climate Hazards Center - U.C. Santa Barbara, 2023). We chose CHIRPS due to its high accuracy with station data, fine spatial resolution (0.05° × 0.05°), and extensive temporal coverage from 1981 to the present.

### 2.2 Temperature

To understand the contribution of temperature to drought, the ERA5 Land temperature dataset (European Centre for Medium-Range Weather Forecasts, 2023) was used in this study. The average monthly gridded dataset was downloaded from the Copernicus Climate Data Store for the study period 2001 to 2021. The original spatial resolution was 0.1x0.1 degrees.

### 2.3 Soil Moisture

We used the ERA5 Land soil moisture dataset (European Centre for Medium-Range Weather Forecasts, 2023) acquired from the Copernicus Climate Data Store for the study period from 2001 to 2021. The monthly data products, with a spatial resolution of 0.1x0.1 degrees, were used for the study. The soil moisture datasets were available for different soil depth levels: first (0–7 cm), second (7–28 cm), third (28–100 cm), and fourth (100–289 cm). Each of these levels has individual importance and characteristics in drought monitoring(Bolten et al., 2010; Sawada, 2018; Sehgal et al., 2017); hence, we employed a weighted method to assign weights to all the levels and created a final single input layer for soil moisture.

### 2.4 NDVI

NDVI serves as a significant indicator of vegetation stress levels. We used monthly NDVI data products from Moderate
Resolution Imaging Spectroradiometer (MODIS) spanning from 2001 to 2021. These data were obtained at a spatial resolution
of 1 km, which we resampled to 0.1 *0.1 degrees to align with other data sources for comprehensive assessment. The MODIS
NDVI dataset was used due to its global coverage and high temporal resolution, making it essential for monitoring vegetation
health, facilitating timely detection, and assessing drought conditions worldwide.

### 2.5 GDIS/EM-DAT

The Global Disaster (GDIS) dataset is a geocoded dataset distributed by SEDAC NASA, which includes geographic
information system (GIS) polygons of affected administrative units by natural disasters (Rosvold and H. Buhaug., 2021). In
this study, the GDIS was used to understand the location of the risk (consequence) of drought, which served as a tool to
comprehensively evaluate the connection between natural hydrometeorological hazards and the socio-economic impact of
drought. The GDIS dataset is based on the EM-DAT dataset. EM-DAT records a natural disaster when it meets any of the
following conditions: 10 or more fatalities, impacts 100 or more individuals, or prompts a declaration of a state of emergency
along with a request for international aid.

To identify any drought events from EM-DAT datasets, their event identifier (disaster no) information was extracted. In EM-
DAT, disaster events are uniquely distinguished by the combination of an eight-digit disaster code and a three-digit country
code, whereas GDIS employs only the eight-digit disaster code. Considering this common ground, GDIS drought events were
shortlisted. In our study period, from 2001 to 2021, a total of 2142 GDIS-based drought events were identified. Since GDIS
lacks drought start and end information, exact yearly start and end information was acquired from EM-DAT. In some cases,
due to the unavailability of monthly details in EM-DAT, we assumed January as the starting month and December as the end
month of the respective event, and further analysis was carried out.





**Table 1. Details of the datasets used for this study.**

| Data | Description and source |
|---|---|
| Rainfall | CHIRPS data (original spatial resolution = 0.05°* 0.05°, further resampled to 0.1°*0.1°) (https://www.chc.ucsb.edu/data/chirps) |
| Temperature | ERA5-LAND data (original spatial resolution = resolution 0.1°*0.1°) (https://cds.climate.copernicus.eu/cdsapp#!/dataset/reanalysis-era5-land-monthly-means) |
| Soil Moisture | ERA5-LAND (original spatial resolution = resolution 0.1°*0.1°) (https://cds.climate.copernicus.eu/cdsapp#!/dataset/reanalysis-era5-land-monthly-means) |
| NDVI | MODIS (original spatial resolution = resolution 1 km* 1 km, further resampled to 0.1°*0.1°) (https://modis.gsfc.nasa.gov/data/dataprod/mod13.php) |
| GDIS | SEDAC NASA, (Original dataset- EMDAT), (Spatial Resolution – Subnational level, Temporal Resolution – Event wise) (https://sedac.ciesin.columbia.edu/data/set/pend-gdis-1960-2018/data) |

## 3. Method

### 3.1 Conventional single-variable-based drought indices

To understand the performance of drought indices in identifying GDIS drought events, we first employed four commonly used individual variable-based traditional drought indices. These included the Rainfall-based SPI (computed using 'SPEI' and 'raster' packages from R statistical software), Temperature-based STI, Soil Moisture-SSI, and Vegetation Stress-related NDVI. Drought assessments were conducted for all these indices on a monthly scale for the period from 2001 to 2021. Here, we incorporated the standardization method (Z score statistics) during index computation of SSI, STI and NDVI to facilitate comparative analysis and noted similarities and differences between the results in identifying GDIS events. Z score values were computed using the following method,

$$Z\ Score = \frac{X - \mu}{\sigma} \tag{1}$$

where,

X = A specific value of a parameter within the set

$\mu$ = long-term mean

$\sigma$ = long-term standard deviation





Further, the results for all four indices were classified and categorized, as shown in Figure 1. Values ranging from zero to -2
or less (negative anomalies) indicate drought conditions, representing mildly dry to extremely dry situations. Conversely,
210    values ranging from zero to 2 or greater (positive anomalies) depict mildly wet to extremely wet conditions, respectively.

| No | Index values | Drought category | Color Code | No | Index values | Drought category | Color Code |
|---|---|---|---|---|---|---|---|
| 1 | 2 or more | Extremely Wet |  | 5 | 0 to −0.99 | Mildly Dry |  |
| 2 | 1.5 to 1.99 | Severely Wet |  | 6 | −1.0 to −1.49 | Moderately Dry |  |
| 3 | 1.0 to 1.49 | Moderately Wet |  | 7 | −1.50 to −1.99 | Severely Dry |  |
| 4 | 0 to 0.99 | Mildly Wet |  | 8 | −2 or less | Extremely Dry |  |

**Figure 1. Drought categories and specific class values used to evaluate the drought indices.**

215    **3.2 Combined Drought Indicator**

Various studies suggested that drought results from variations in multiple agro-climatological settings rather than just a single
variable. The socio-economic repercussions of drought could occur due to combinations of multiple factors rather than a single
one. Hence, in this study, we developed a new combined drought indicator to assess droughts by considering multiple
agricultural and climatological variables. We then checked and compared its association with GDIS events in the method
220    described in Section 3.3. The CDI was developed using two agricultural parameters (Soil Moisture, NDVI) and two
climatological parameters (Rainfall, Temperature). The CDI was generated as the weighted average of four independent
drought indices shown in Section 3.1. The Principal Component Analysis (PCA) technique was used to assign the weights to
all four input indices. PCA has been widely used in atmospheric and hydrological studies to describe dominant patterns in the
data. Through this method, new orthogonal (independent of each other) variables, i.e., P.C.s, were constructed using linear
225    combinations of the variables without losing much information. It is understood that the highest variability of input data could
be reflected in the first P.C. (PC1). Therefore, the PC1 values were used for the further steps. In PCA, the principal components
(P.C.s) are found to maximize the variance of each P.C., and the total of the squared loadings (eigenvectors) equals one. The
squared loadings show how much each variable contributes, which is figured out using the eigenvectors. Using this approach,
we generated 48 spatially weighted maps for each month (totaling 12 months) and all four input variables (SPI, STI, SSI, and
230    NDVI). The percentage weights obtained from PCA were then used to assess the CDI using the formula below:

$$Weighted\ CDI(i,j) = WtSPI(i) \times [SPI](i,j) + WtSTI(i) \times [STI](i,j) + WtSSI(i) \times [SSI](i,j) + WtNDVI(i) \times [NDVI](i,j)$$

(2)

Where WtSPI, WtSTI, WtSSI, and WtNDVI represent the weights for the ith month (January to December) for their respective
parameters. These weights were multiplied by the individual index values for the jth year and ith month, and then all the values





were added to obtain the CDI results for the ith month and jth year. Finally, all the CDI results were normalized for comparative
analysis. This normalization process was carried out using the following formula:

$$CDI(i,j) = \frac{weighted\ CDI(i,j)}{\delta(i)} \tag{3}$$

Where, CDI(i,j) represents the combined drought anomaly for the ith month and jth year, while $\delta(i)$ denotes the standard
deviation for the ith month across all years (2001 to 2021). The CDI results were showcased using the same color scheme and
drought categories, as depicted in Figure 1. Figure 2 illustrates the schematic flowchart outlining the process for computing
PCA-based CDI.

In the later stage, the same process as that for individual indices (SPI, STI, SSI, NDVI) outlined in section 3.1 was followed
to assess the association between CDI results and GDIS drought events. Similarities and differences observed by CDI in
identifying GDIS events compared to SPI, STI, SSI, and NDVI across various temporal scales and with different criteria were
examined. Detailed explanations of these results are provided in the following section.

### 3.3 Evaluation of the drought indices by GDIS

We assessed the number of events among the shortlisted GDIS drought events (see Section 2.5) that were detectable by each
drought index. This analysis suggests how well each index detects socio-economic stress during droughts. This analysis was
performed using ArcMap 10.8. The GDIS polygons were overlaid onto individual drought indices (SPI, STI, SSI, NDVI, and
CDI). Subsequently, raster-based information was extracted for all 2142 events from each respective data layer. During this
process, event details such as location and start and end dates were obtained from the GDIS and EM-DAT data. The index
values for each specific drought event were collected from the raster layers of the corresponding drought indicator. For
instance, if a GDIS event occurred in Bihar, India, from March to December 2012, the raster information for SPI, STI, SSI,
NDVI, and CDI for Bihar during that period was extracted.

The resulting geodatabase tables were then analyzed to determine whether the index values were consistent with the GDIS
data. Figure 1 illustrates that values below -1 indicate moderately to extremely dry conditions, which has been widely accepted
in previous works. Therefore, this criterion was used to assess the consistency of the drought indices with GDIS. For example,
in the case of Bihar, India, during the GDIS drought period from March to December 2012, if any month within that timeframe
shows a value of -1 or less than -1 for SPI, STI, SSI, NDVI, or CDI in the extracted geodatabase table, then that particular
index would be considered consistent with the GDIS for that specific event. Similarly, all 2142 drought events in GDIS from
2001-2021 were examined for each of the four indices. Additionally, for a detailed comparative analysis, the threshold criteria
were adjusted from -1 to 0, and a similar analysis was conducted. This means that if any month within the specific time frame





of the GDIS event exhibits a value below 0 in the index data, then that index will be considered consistent with GDIS for that
particular event.

The occurrence of GDIS events (socio-economic repercussions) may not solely originate from conditions in a particular month
but could also reflect agro-climatological stress conditions from the preceding one, two, or three months. Hence, we also
analyzed the GDIS events and their association with all five indices by adjusting the duration of event occurrences. We
conducted additional analysis by extending the data extraction timeframe for all five indices. For example, in Bihar, India, for
the GDIS drought event from March to December 2012, we not only extracted index data for March to December 2012 but
also for one month before the event (February), two months prior (January), and three months prior (December 2011) for SPI,
STI, SSI, NDVI, and CDI and checked if the index anomalies are consistent to GDIS or not in any of these durations.

Research suggests that the shorter-duration dry periods have significant importance for various socio-economic applications,
such as water availability and energy sources in adjacent months (Christian et al., 2021, 2024; Mukherjee and Mishra, 2022).
Hence, during the shortlisting process of the 2142 GDIS drought events from the entire natural disaster dataset, the initial
criteria were to consider all the events, including short droughts (droughts lasting more than one month or up to two months).
Consequently, in the later stage of this research, to comprehend the importance and differences in the selection of timescales,
the initial criteria were slightly adjusted. Instead of considering all events, only events lasting more than or equal to two months
were examined. This criterion was established to mitigate the impact of short-term changes and biases in results caused by
short droughts or smaller dry periods. After the implementation of these criteria, we got 1641 GDIS drought events.

To summarize, the comparative analysis of GDIS with all the conventional indices (SPI, STI, SSI, and NDVI) and CDI was
carried out based on 1) two types of drought events—those lasting greater than or equal to two months and all events, including
those with shorter durations (i.e., less than two months). 2). During this process, four temporal data scales were considered:
the actual event period, data from one month prior to the event, data from two months prior to the event, and data from three
months prior to the event. 3) For all the indices, events were classified as drought if the values were below 0 (in the first case)
and below -1 (in the second case); the remaining events were marked as non-drought events.



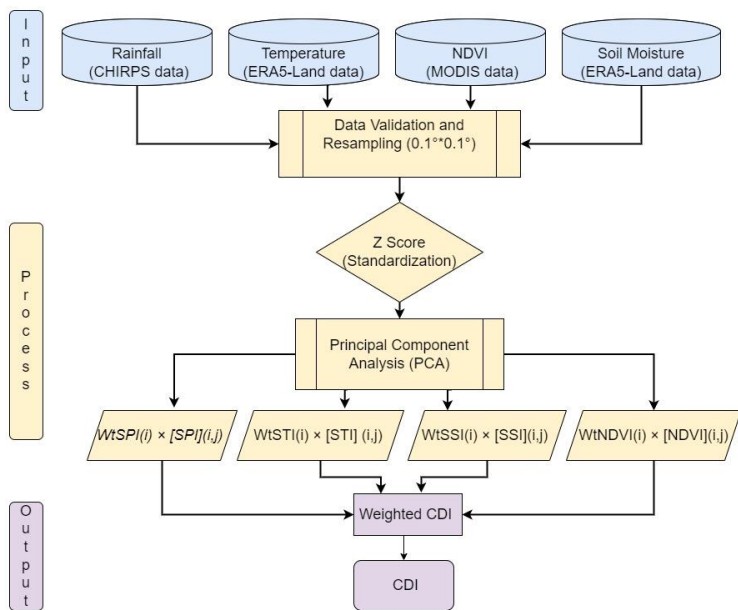


**Figure 2. The general flow of method followed to compute CDI-based droughts. Here, i represents the month from January to December, and j indicates the years from 2001 to 2021.**

## 4. Results

### 4.1 Global Drought Frequency Analysis by GDIS Events

The GDIS dataset contains location data for 39,953 events globally, including various natural disasters such as extreme temperatures, floods, mass movements, storms, droughts, and earthquakes. Among these, 2,142 events were explicitly identified and classified as drought events. Figure 3(a) illustrates the global spatial distribution of drought frequencies of all 2,142 GDIS drought events. The highest number of these drought events were recorded in Africa (779) and Asia (710), followed by the Americas (477), Europe (106), and Oceania (70). Analyzing drought frequencies at a finer scale, Djibouti in

Africa reported the highest frequency of GDIS drought events (8 occurrences), followed by Ethiopia and Kenya (7 occurrences each) and Somalia (6 occurrences) (Figure 3b.). In the Americas, Bolivia recorded the highest number of GDIS drought events in South America, while Honduras had the highest frequency in Central America (7 occurrences). In North America, Kansas County in the USA documented the highest number of GDIS drought events (4 occurrences), followed by California, Arizona, and Illinois with three occurrences each (Figure 3c.). In Asia, the Nei Mongol administrative unit in China documented the

highest frequency of GDIS drought events (7 occurrences). This is followed by several administrative units in Thailand, including Chaiyaphum, Loei, Nan, and Kalasin, which each recorded a high frequency of GDIS drought events (6 occurrences), indicating a significant vulnerability to drought in these regions. Additional regions in China (such as Yunnan, Shanxi, and



Hebei), India (Maharashtra and Andhra Pradesh), and Cambodia (Pursat and Kampong Speu) documented moderate GDIS drought frequencies (3 to 5 occurrences) (Figure 3d.).


On the temporal scale, parts of Africa and South America experienced the highest extended drought events. The drought over Burundi, Africa, lasted continuously for two years from 2004-2006, whereas the drought in South America, over Honduras, lasted for around a year in 2009. Some of these events were very harshly socio-economically devastating. EM-DAT data showed that, in a major event in Africa, nearly 2 million people were affected, whereas in Thailand, around 3 million people

faced the impacts of drought.

It is also noteworthy that areas with low population density, like western Australia or central and western Russia, did not experience any GDIS events. In contrast, higher population areas were more prone to the GDIS events, making population density a key factor in socio-economic stress. Additionally, it was observed that areas with smaller spatial scales are less likely

to experience GDIS events compared to larger areas.

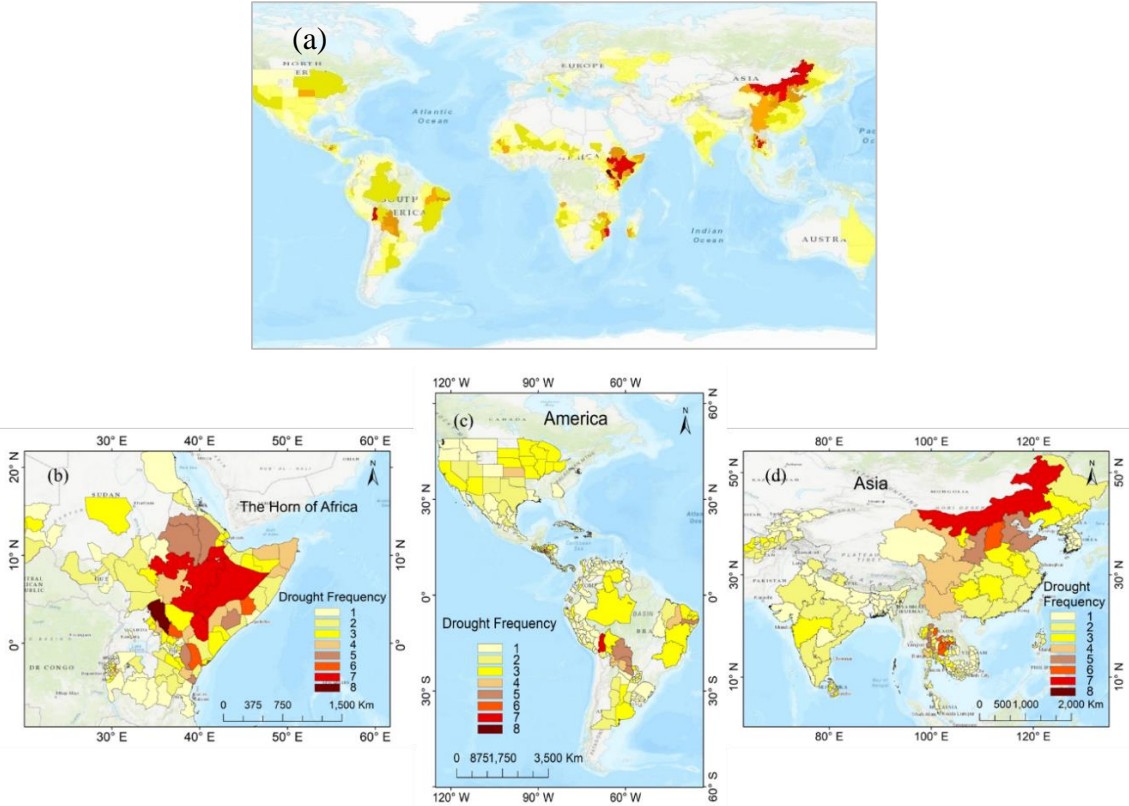

**Figure 3. Spatial Distribution of GDIS drought frequencies (a) Global scale (b) East Africa (c) America and (d) Asia. The drought frequencies range from one to eight, represented by shades ranging from light yellow to dark brown, respectively.**





## 4.2 Performance of CDI in detecting GDIS events.

In the computation of the CDI, one of the initial steps involves assigning weights to all input variables using the PCA method. Figure 4 shows the pixel-based weights for each variable for the sample month of April. Similar weights were computed for all twelve months across four input variables. In April, the highest weights were assigned to soil moisture, indicating its greater contribution to the development of CDI compared to NDVI, LST, and rainfall. Lower weights were assigned to LST, suggesting a weaker correlation with other variables and a lesser role in CDI formation for April. Significant variations in these weights have been observed across different global regions, reflecting seasonal dynamics. In the tropical areas of Southeast Asia and parts of Africa, rainfall and soil moisture exhibit higher weights during the monsoon seasons from June to September, while NDVI becomes more prominent in the subsequent months. Whereas, in parts of North America, NDVI shows higher weights in September and October, followed by soil moisture and rainfall.

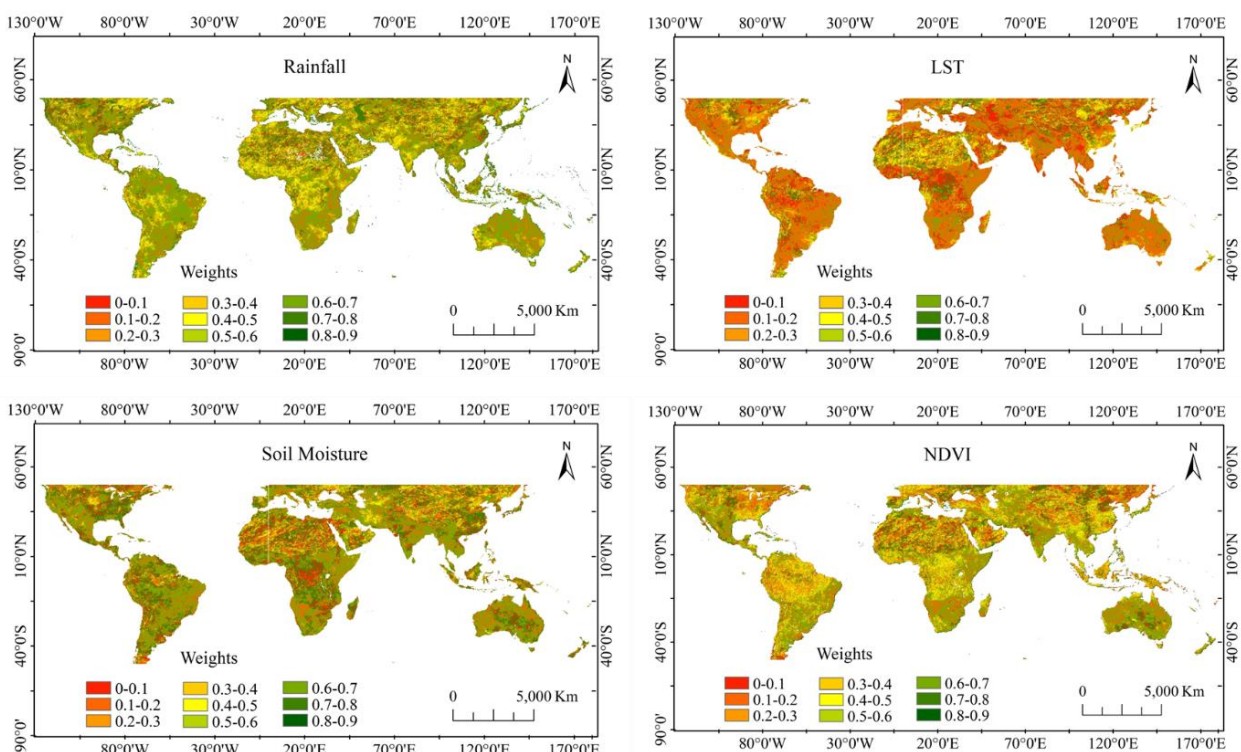

**Figure 4. Pixel-based weights for four input variables of CDI (rainfall, LST, soil moisture, and NDVI) computed for the month of April using the PCA method. The weights range from 0 to 1, with colors ranging from dark red (lower weights) to dark green (higher weights).**

The CDI maps helped to identify different categories of droughts between 2001 and 2021 over various regions of the globe (Appendix 1). Severe to extreme drought conditions were observed over South African countries such as Malawi and Zambia, as well as the Horn of Africa, including Ethiopia, Kenya, and Somalia, during 2015-16. In the United States, the years 2007-





08 and 2012 were marked as severe to extreme drought years, affecting the southeastern states and the Midwest, respectively. During November 2009, the CDI noted one of the most destructive drought periods in northern and western China. The year

2015 was marked by the CDI as a severe drought year in India, affecting 52% to 67% of the region and causing massive agrarian stress. Additionally, the CDI data indicated that during early 2019, areas of Australia, particularly Queensland and New South Wales, experienced some of the most severe drought conditions recorded in recent years. Further, overlapping the GDIS polygons on these CDI results helped to understand the association between CDI and GDIS drought events.

Figure 5 demonstrates the performance of CDI in detecting GDIS events. Figure 5a shows CDI for North and Central America in June 2015 overlaid with black polygons representing GDIS data for the same period. Similarly, Figures 5b and 5c showcase the CDI maps for Africa in June 2009 and South America in August 2009, respectively, aligning with periods of reported GDIS events with their corresponding polygons. In these examples, it is clearly seen that GDIS polygons are exactly aligned with severe to extreme drought areas, as noticed through the base maps of CDI. Likewise, out of a total of 2142 GDIS global

drought events (with AEP), CDI can detect 1885 drought events when the drought criteria were set to -1 or less and 2117 when the criteria were set to 0 or less. After adjusting the criteria to consider drought events lasting two months or longer, the total event count was reduced to 1641 GDIS events. Under this criterion, CDI identified 1550 GDIS events when the drought threshold was set to -1 and 1635 when the threshold was set to 0. Adjusting the criteria and thresholds more rigorously (one month prior to the end of GDIS events, two months prior to the end of GDIS events, and three months prior to the end of GDIS

events) resulted in a stronger association between GDIS and CDI, with percentages ranging from 91% to 100% (Table 2).

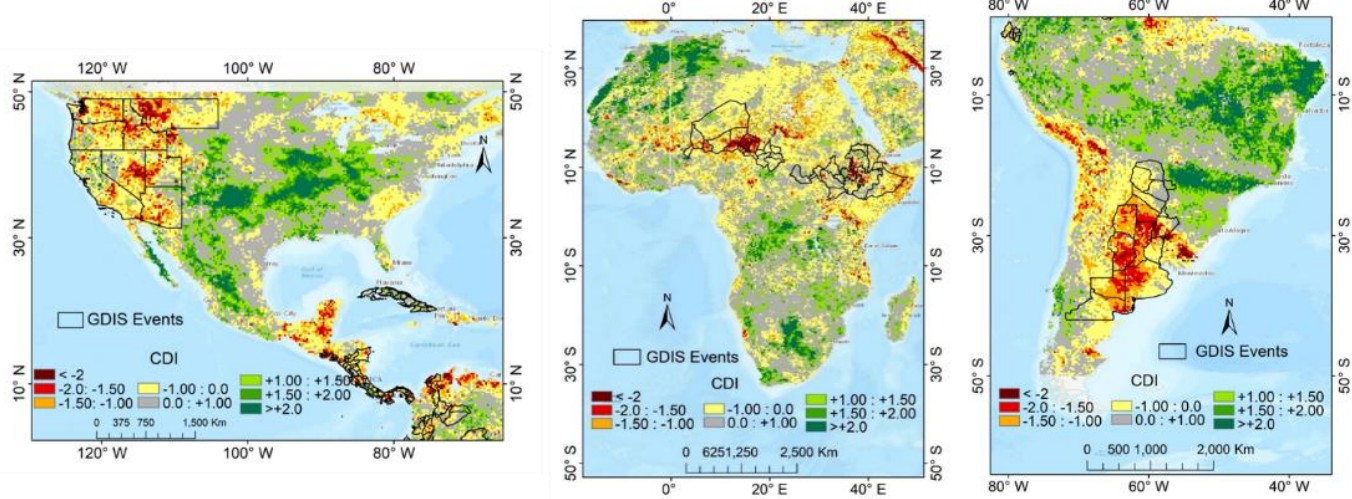

**Figure 5. Assessment of Drought Using CDI with Overlay of GDIS Events over North and Central America (June 2015), Africa (June 2009), and South America (August 2009). The black polygons represent GIS polygons from GDIS, indicating drought-affected administrative units based on GDIS data. The base maps display CDI results for the respective GDIS drought months, ranging from**

**dark brown (indicating extreme dry conditions) to dark green (indicating extreme wet conditions). The alignment of GDIS polygons with droughts detected by CDI demonstrates the CDI's capability to accurately identify GDIS droughts during the respective periods.**





**Table 2 Performance of CDI in Detecting GDIS Events Using Multiple Criteria**

| | Total Event | Drought Criteria | Actual Event Period (AEP) | | One Month Prior + AEP | | Two Month's Prior + AEP | | Three Month's Prior + AEP | |
|---|---|---|---|---|---|---|---|---|---|---|
| | | | -1 | 0 | -1 | 0 | -1 | 0 | -1 | 0 |
| **Including Short Drought** | 2142 | Observed | 1885 | 2117 | 1954 | 2130 | 2010 | 2137 | 2042 | 2142 |
| | | % | 88.00 | 98.83 | 91.22 | 99.44 | 93.84 | 99.77 | 95.33 | 100 |
| | | Not Observed | 257 | 25 | 188 | 12 | 132 | 5 | 100 | 0 |
| | | % | 12 | 1.17 | 8.78 | 0.56 | 6.16 | 0.23 | 4.67 | 0 |
| **No Short Drought (event >=2 months)** | 1641 | Observed | 1550 | 1635 | 1573 | 1637 | 1587 | 1637 | 1589 | 1641 |
| | | % | 94.45 | 99.63 | 95.86 | 99.76 | 96.71 | 99.76 | 96.83 | 100 |
| | | Not Observed | 91 | 6 | 68 | 4 | 54 | 4 | 52 | 0 |
| | | % | 5.55 | 0.37 | 4.14 | 0.24 | 3.29 | 0.24 | 3.17 | 0 |


Figure 6 illustrates the performance of CDI in detecting GDIS drought events across different regions (Appendix 2 represents the performance of traditional indices). In this analysis, the drought identification criteria were set as all GDIS events, including short-duration cases, with a drought threshold set at -1 or less. Out of a total of 2,142 GDIS events, the highest number was observed in Africa, with 779 events, of which CDI successfully detected 678. Following Africa, South and Southeast Asia experienced 613 GDIS drought events, with CDI accurately identifying 526 of these. In South and Central America, there were 434 GDIS drought events, and CDI identified 387 of them. East Asia and the Pacific saw 144 GDIS events, with CDI detecting 138. Europe and Central Asia, as well as Northern America, had the fewest GDIS events. CDI detected 111 events in Europe and Central Asia and all 53 events in Northern America. These findings indicate that CDI exhibits high accuracy in regions with larger subnational scales compared to areas with smaller scales. The 100% detection rate of GDIS droughts in Northern America by CDI may be attributed to the extensive areal extent of the region. In line with our results, a previous study by Kageyama and Sawada (2022)also showed that developed countries are easier to detect. These countries did not suffer from droughts without strong climatological hazard signals, which enhances the capability of our climatology-based hazard index. The vast subnational scale of Northern America includes a greater number of CDI image pixels, which likely enhances the correlation with GDIS events.

Despite the higher association of CDI in detecting GDIS events, there were a few instances (e.g., Burundi, and parts of Thailand) where CDI failed to capture GDIS events. The smaller spatial extents (subnational scales in Southeast Asia or parts of Africa) could be one reason for this discrepancy. Additionally, these regions might be experiencing other types of stress beyond agro-climatological factors (as indicated by CDI) that contribute to the GDIS events. It was also found that CDI identified stress conditions in some locations that were not reflected in GDIS. For instance, CDI detected severe drought in South Argentina during 2014-15 and Namibia in 2013. Additionally, CDI identified drought conditions over parts of Europe in 2018 that weren't reflected in GDIS data. These disparities may arise from adaptation techniques or practices implemented




in these regions, effectively managing agro-climatic extremes or hazards without significantly impacting local socio-economic

conditions.

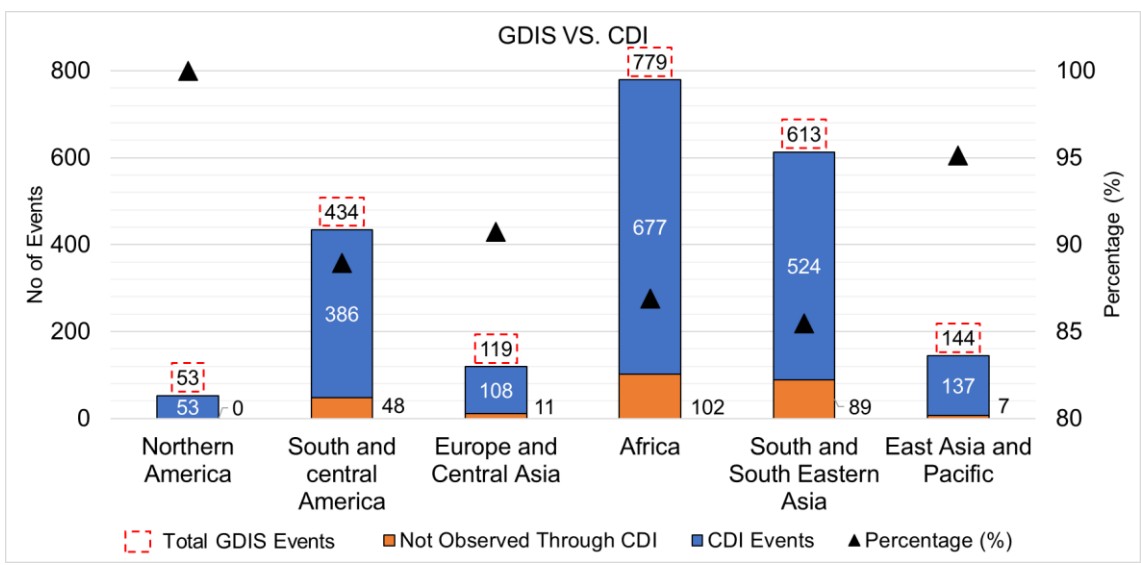

**Figure 6. Performance of CDI in Detecting GDIS Events Across Different Geographic Regions. A total of 2142 drought events were identified from GDIS. CDI successfully detected events marked in blue, with counts indicated by red dotted boxes. Events missed**
**by CDI are represented in orange. The black triangles indicate the percentage of GDIS events captured by CDI.**

**4.3 Comparative Assessment of CDI and individual parameter based traditional indices**

Figure 7 presents a comparative analysis of multiple drought indices, including CDI, in detecting GDIS events using various criteria. When considering the initial criteria (Figure 7 a), which covers the actual event period of GDIS and includes all events,
including short-duration droughts, a total of 2142 events were identified in GDIS. Among these, CDI exhibited the highest detection rate with 1885 events, followed by NDVI with 1867 events, SSI with 1170 events, SPI with 1740 events, and STI with 1650 events. Here, the criterion was set to consider all events, including short-term drought occurrences, with the drought threshold set to -1 or lower. The number of the GDIS drought events captured by each index can easily be increased by the lower threshold of drought identification (Figure 7(a,b,c,d), subplot 2), although it inevitably increases the false alarms. This
association remained consistent when the drought criteria were further refined to exclude events shorter than two months (Figure 7(a,b,c,d), subplot 2). It's also notable that the agricultural indices (NDVI and SSI) exhibit greater efficacy in identifying GDIS drought events compared to meteorological indices (SPI and STI). This suggests that they may offer a more accurate representation of socio-economic conditions. Through this analysis, we observed that NDVI performs better in regions with diverse vegetation cover and seasonal variability, such as the Indian subcontinent and South America (Appendix 3). In



contrast, SSI detected more drought events than NDVI in semi-arid areas like Central Asia (e.g., Kyrgyzstan, Afghanistan), where NDVI is less informative due to sparse vegetation.

When considering an alternative criterion that includes drought data spanning two months prior to the onset of the GDIS event until its end date, with the exclusion of short-term drought occurrences, a total of 1641 GDIS events were observed. Out of

these 1641 events, CDI demonstrated the highest efficacy by detecting 1587 events. Subsequently, NDVI identified 1574 GDIS events, while SSI detected 1580 events. SPI observed 1550 events, and STI represented 1510 events, showcasing its lesser capability in detecting GDIS events. It is observed that, across most criteria (Figure 7 b,c,d), CDI demonstrated a superior capability in identifying GDIS events compared to other indices. It is also observed that, across most criteria, the CDI has a greater potential to detect GDIS droughts compared to traditional drought indices based on single input variables (Appendix

425 4).

However, there were three instances where CDI lagged other indices, primarily NDVI or SSI, in detecting GDIS events. In the first scenario (Figure 7b, criterion: one month prior to AEP and including short droughts), CDI detected 1954 GDIS events, while NDVI identified 1983 GDIS events. Similarly, in the second scenario (Figure 7b, criterion: two months prior to AEP and including short droughts), NDVI maintained its superiority in detecting GDIS events, observing 2028 events compared to

CDI's detection of 2010 events. In the third scenario (Figure 7c, criterion: three months prior to AEP and no short droughts considered), although CDI detected 1589 GDIS events, SSI outperformed by observing 1610 GDIS events. These discrepancies may be caused by CDI giving more weight to input parameters such as SPI and STI, influenced by regional variability, local environmental conditions, and land cover patterns during these event periods.

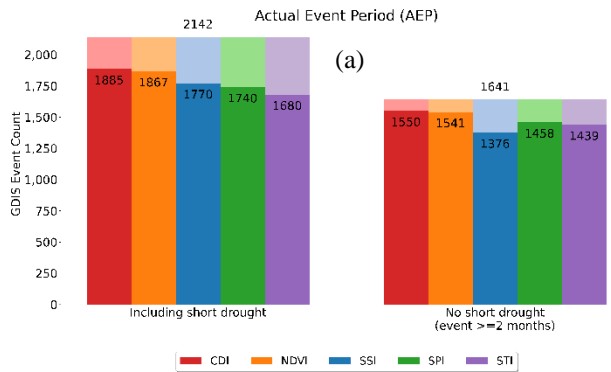
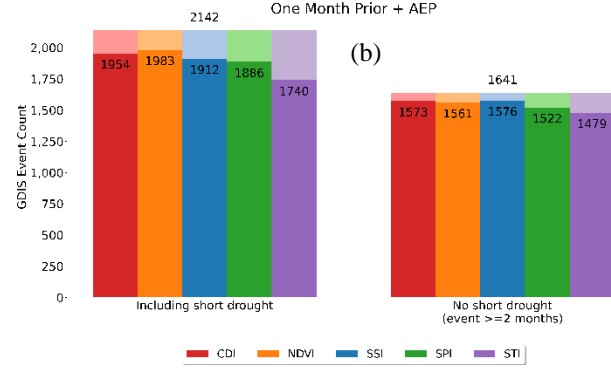





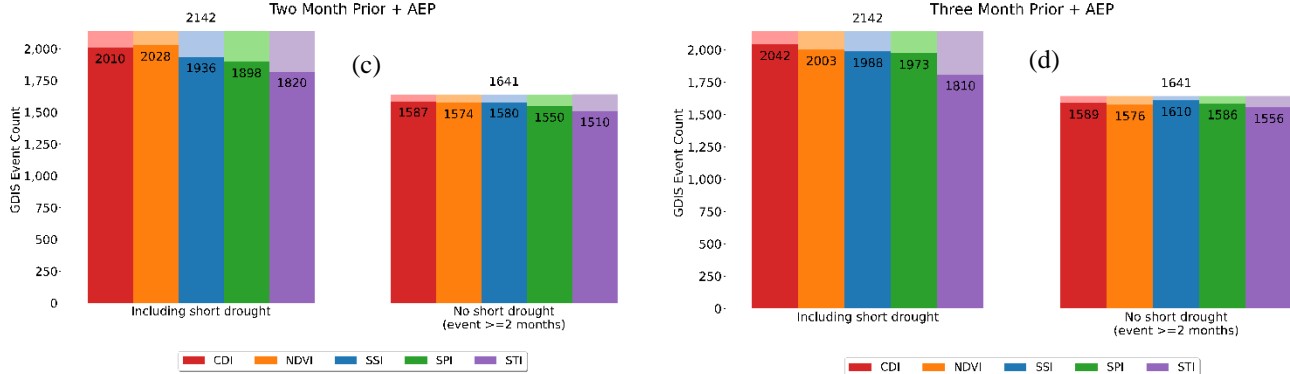

**Figure 7. Comparative Assessment of CDI vs Traditional Drought Indices in Detecting GDIS Droughts (Drought Identification Threshold: -1 or less). The first subplots of a, b, c, and d illustrate the performance of indices in detecting droughts from a total of 2142 GDIS drought events (considering short drought events). The second subplot of a, b, c, and d shows the performance of indices in detecting GDIS drought events when short drought events are excluded, resulting in a total count of 1641. Actual Event Period (AEP) is the exact period when the GDIS event occurred, for which index data has been considered. One month, two months, and three months+ AEP are the respective previous months relative to GDIS events plus the actual event period from which index data was considered to understand the lag effect in GDIS event occurrences.**

One of the reasons for the superiority of CDI in detecting GDIS drought events is its flexibility.CDI can be tailored to suit local or regional contexts, taking into account the unique characteristics represented by specific traditional indices or combinations of multiple indices. Figure 8 provides examples of such cases, where one or more individual indices represented GDIS drought events, allowing CDI to also detect those GDIS events. Figure 8 (a) shows one of the event cases over Busoni, Burundi, from Africa, where the GDIS event was observed from April 2014 to September 2014. In this case, the lines for NDVI, SPI or STI didn't cross the drought threshold (-1) during the particular period, hence couldn't detect the GDIS event. SSI could detect this drought, and CDI has a relatively large weight on SSI, so CDI could also detect it. The average rainfall is low with small variability over this region in this period, making SPI not detecting this event, whereas the common hot and semi-arid climate of this region would not have helped STI and NDVI to detect drought over Burundi. The decreased availability of groundwater or surface water in this area likely resulted in reduced soil moisture supply, contributing to SSI drought. In the second example (Figure 8b), a GDIS drought event was observed over Gaya, Bihar, India, from May 2009 to September 2009. This region is located along the banks of the Ganga River, which likely provided ample irrigation. Consequently, SSI and NDVI did not indicate a drought in this area. However, the insufficient rainfall over Gaya during this period caused socio-economic stress, which was detected by both SPI and CDI.

Figure 8(c) shows the performance of various indices over Holguin, Cuba, where a GDIS event was reported from August 2004 to March 2005. In this case, STI and SPI did not detect the drought, whereas NDVI and SSI indicated drought conditions, leading to the CDI's detection of the drought as well. Despite non-drought conditions in precipitation and temperature, agricultural stress factors such as poor vegetation health or subsurface water depletion could have contributed to this drought, which was effectively captured by NDVI, SSI, CDI, and GDIS. In the fourth example (Figure 8d), a GDIS event was observed



over Nebraska, USA, from June 2012 to November 2012. All five indices (NDVI, SSI, STI, SPI, and CDI) detected this event, highlighting the extreme drought severity during this period in the USA.

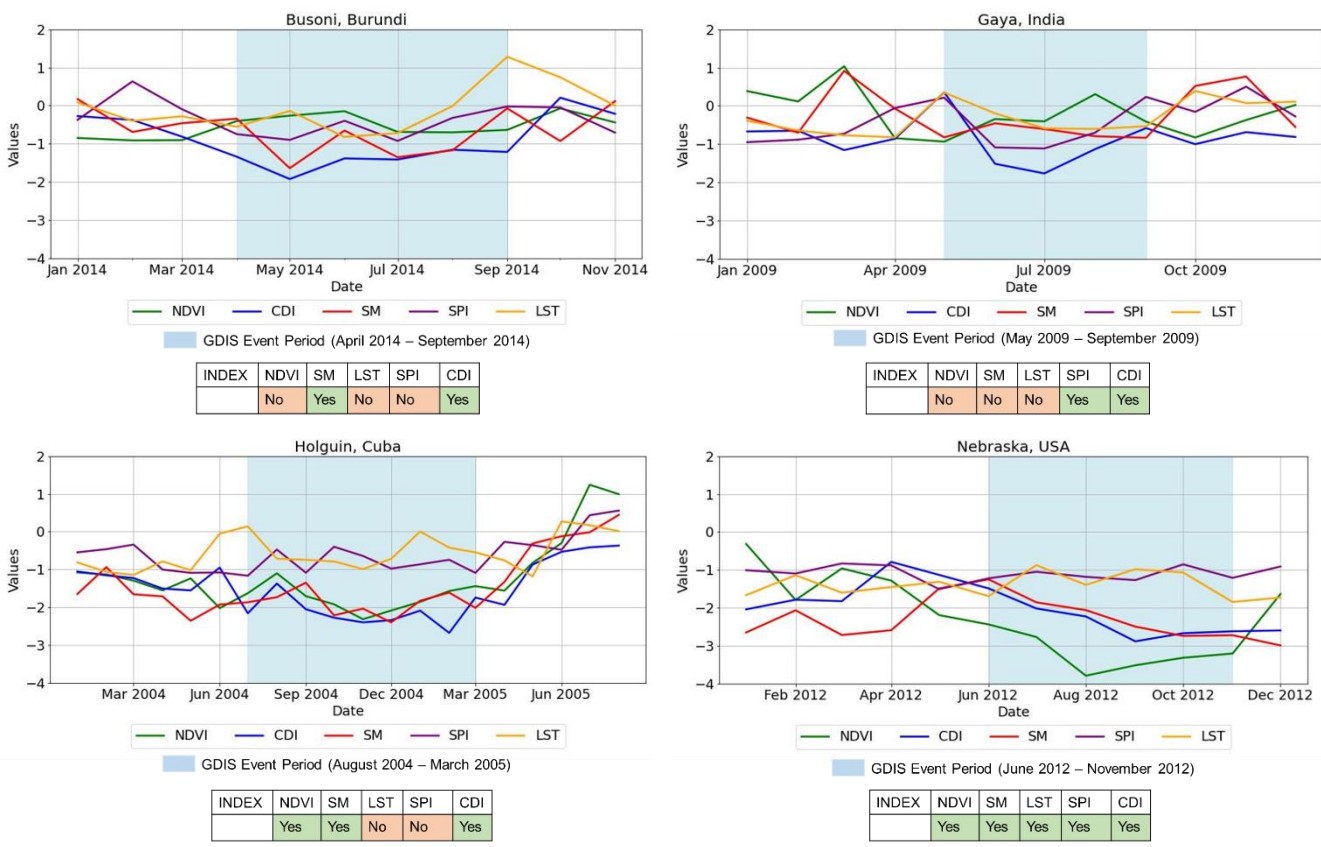

**Figure 8. Performance of CDI, SPI, STI, SSI, and NDVI in Detecting GDIS Events Over Burundi, Gaya, Holguin, and Nebraska. The blue boxes denote the duration of GDIS events, while colored line graphs represent the index values of NDVI (green), CDI (blue), SSI (red), SPI (purple), and STI (yellow). If the line of any index value crosses the drought threshold of -1 during the specified GDIS event period (blue box), that index is considered capable of identifying the GDIS drought event.**

In general, strong correlations were observed between CDI and SSI, followed by NDVI, SPI, and STI among the four single input-based traditional indices. CDI was highly associated with SSI over the Indian subcontinent, Australia, and South America. However, these correlations exhibited spatial and temporal variability. Monthly assessments revealed significant seasonal variations in the correlations between CDI and other indices. During monsoon months (June, July, August, and September), CDI exhibited a higher correlation with SPI over the Indian subcontinent compared to non-monsoon months. Similarly, in arid regions of Africa, strong correlations between NDVI, SSI, and CDI were observed during rainy months (June to September), which diminished during dry months. As a sample example, Figure 9 depicts the spatial correlation between





CDI and other indices in the month of April. On average, NDVI exhibited a higher correlation with CDI, particularly in northern America, southern parts of Africa, and the Indian subcontinent. Following NDVI, SSI and SPI demonstrated stronger

associations with CDI. However, in April, CDI exhibited a lower correlation with STI. Parts of South America, South Africa, Australia, and the Indian subcontinent even displayed a negative correlation between CDI and STI, indicating worsened or more severe drought impacts during this period.

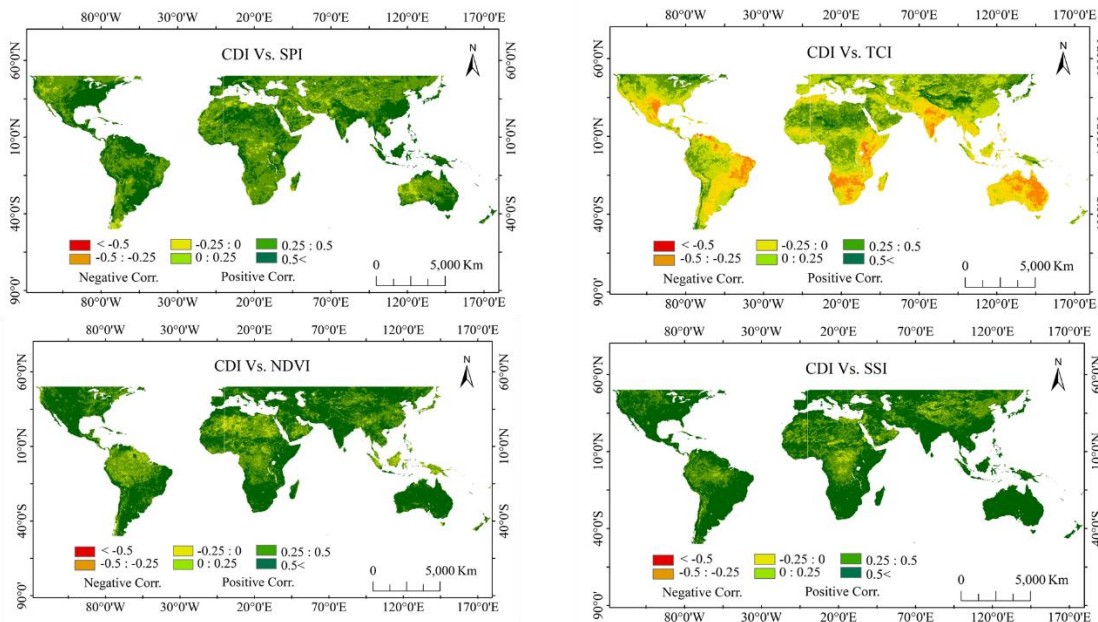

**Figure 9. Spatial Correlation Between CDI and Single Input-Based Traditional Indices (Sample Example for April). Negative correlation is represented by colors ranging from yellow to red, while positive correlation is depicted in shades from light green to dark green.**

## 5. Discussion

One major challenge in drought studies is assessing the aftereffects and propagation of specific drought events on society and

their long-term impacts. Some researchers have attempted to shed light on this issue, using proxies to show the socio-economic effects. Noel et al. (2020) compared weekly hazard maps with socio-economic impacts across the USA, and Bachmair et al. (2016) described thresholds for hydrometeorological droughts to understand the socio-economic impacts of droughts in Germany and the U.K. However, these studies were region-specific and insufficient to provide a global perspective. Our study addressed this research gap with the help of GDIS data. The use of GDIS data in this study allowed us to obtain precise

subnational information on disaster locations, and instead of relying on proxies, this dataset provided direct on-ground impact information. Consistent with the findings of Kageyama and Sawada (2022), our study also observed that Sub-Saharan Africa





and South Asia experienced a higher number of GDIS drought events, highlighting the high vulnerability of these regions to droughts and their socio-economic repercussions.

Previous studies, whether regional or global, often relied on single-parameter-based indices for drought monitoring, each of which has limitations for understanding drought. In the case of SPI, its effectiveness depends on the chosen time scale (e.g., SPI 1, SPI 3, etc.). In regions with distinct wet and dry seasons, SPI can sometimes misrepresent actual drought conditions. For example, a short-term SPI might indicate drought during a naturally dry season, even if overall water availability is within normal limits for the year. Regarding NDVI, results may vary depending on the type of vegetation, as different plants respond
differently to drought stress, which may not always be captured by NDVI. Additionally, particularly in places with abundant rainfall, NDVI may not be able to differentiate between dense vegetation and healthy vegetation. Soil moisture index has limitations in its spatial coverage and the variability of its levels over time, which can respond differently to changing conditions. Hence, the development of CDI in our study and its analysis on a global scale marks a significant departure from previous studies that focused on single input parameters. By integrating multiple parameters for the analysis, CDI offers a
more comprehensive understanding of drought conditions, transcending the limitations inherent in single-parameter approaches. Bayissa et al. (2022), Jiao et al. (2019), and Kulkarni et al. (2020), etc. employed a comprehensive approach, emphasizing the importance of CDI in their regional drought monitoring studies over Sri Lanka, India, and the USA, respectively. However, these studies were limited to regional analyses. Hence, in this study, taking a step ahead, we analyzed droughts using the CDI technique at the subnational level on a global scale.


In this study, the computation of CDI-based droughts involved implementing the PCA technique to assign weights to each input parameter. In contrast, previous researchers (Kulkarni et al., 2020b; Thomas et al., 2016, etc.) relied on the expert judgment method or copula method (Shah and Mishra, 2020; Tosunoglu and Can, 2016)to assign weights for the input parameters. The expert judgment method is highly subjective, and weight assignments can fluctuate based on individual
opinions. Furthermore, the Copula method is more susceptible to outliers, potentially leading to biased weight assessments compared to PCA. Our study addressed these limitations by employing the PCA technique. PCA enhances the assessment of drought severity by objectively identifying influential variables and capturing data patterns, thereby enhancing the accuracy and reliability of our findings.

The comparative analysis between the CDI and other individual parameter-based indices in identifying GDIS events clearly indicates that the CDI method has the highest potential for detecting GDIS events and a stronger association with socio-economic conditions. Inconsistencies between the CDI and other indicators may stem from the different mechanisms these indices use to detect droughts versus the actual ground conditions. One such inconsistency was observed in the Horn of Africa, where the CDI and other indices identified a GDIS drought event, but the SPI did not. This discrepancy could be due to the
less pronounced anomalies in the SPI for this region. The normal precipitation levels in the Horn of Africa are already low,





which might not always indicate droughts. However, decreased soil moisture, decertified land, or soil types with low moisture retention would result in drought conditions, as indicated by the SSI, STI, NDVI, and CDI. Another example is from North Argentina (South America), where the SSI failed to detect a drought that was identified by other indices. This disparity might be due to the presence of the Paraná River, the second-largest river in South America, which provides a significant source of

soil moisture. Therefore, the SSI might not reflect drought conditions, even though less rainfall or higher temperatures could cause drought and socio-economic stress, as shown by the GDIS. A further example is from Ballia (Uttar Pradesh, India), where the SPI and SSI did not detect a GDIS drought, but the NDVI and CDI did. Ballia is near the Ganga River, which supplies ample soil moisture to the surrounding areas. However, the NDVI is particularly effective at identifying droughts in highly vegetated areas due to its sensitivity to precipitation, and soil moisture and its inverse correlation with temperature.

Thus, the drought was detected by the NDVI and CDI.

Through this study, we observed that the relationship between hazard (CDI) and socio-economic impacts (GDIS) is significantly more complex than initially anticipated. The relationship between CDI and GDIS, as well as their vulnerability to drought, varies markedly between developed and less developed regions. Despite some areas in North America and Europe

having lower CDI values (less than or equal to -1.5), these regions have not exhibited corresponding GDIS (socio-economic repercussions of droughts) droughts. Conversely, relatively higher CDI values (less than or equal to -1.0 or even -0.5 in some cases) over South Africa or parts of Asia and South America have led to noticeable GDIS events (socio-economic impacts). Meaning that, the threshold values of drought impacts (CDI) differ across regions. These findings align with previous studies (Tanoue et al., 2016; Tschumi and Zscheischler, 2020), which represent the behavior of developed versus developing nations,

showing that despite experiencing higher climatic anomalies, developed nations are less likely to be socio-economically affected by disasters compared to developing countries. To support this notion, Tanoue et al. (2016) mentioned that vulnerability is associated with the gross domestic product (GDP) of a region, with higher GDP (Europe, North America) indicating less vulnerability and lower GDP (South Africa, South America, South Asia) indicating greater vulnerability. Similarly, Birkmann et al. (2022), Chen et al. (2020), and Lavell et al. (2012) claimed that developed regions, with their robust

infrastructure, diversified economies, and social safety nets, are better equipped to cope with drought impacts compared to less developed regions with limited resources and institutional capacities. For instance, developed countries may have access to advanced irrigation systems and drought insurance schemes, whereas less developed countries often rely heavily on rainfed agriculture and face greater challenges in mitigating drought impacts. By understanding the association between CDI and GDIS, our study helped to highlight the disparities and complex dynamics of drought hazard and vulnerability between

developed and developing nations. These insights could be used to inform area-specific policy implementation and management practices.

Although 96% of GDIS events aligned with CDI, there were a few GDIS events that CDI could not capture. This indicates that there was socio-economic stress during these periods, but it was not due to hydro-climatological drought hazards. Other





factors must have contributed to this stress. One such event was observed in Burundi (Africa) from June 2001 to December 2001 and in Uganda and Swaziland in southern Africa. Another inconsistent GDIS event was noticed in Peru (South America) in 2010 and Thailand in 2005, when no significant CDI anomalies were observed, yet these periods were noted in the GDIS dataset. These observations demand a more detailed analysis of these events and an understanding of the regional characteristics that might have led to these discrepancies.


The association between the CDI and the GDIS is a significant takeaway from our study, as it directly addresses the gap in understanding actual drought hazards and their socio-economic impacts. However, there are some limitations to this study. One of the primary limitations is the incompleteness of GDIS data. Although GDIS provides valuable information on impacts, the dataset only covers 60% of events registered in EM-DAT at the subnational level. Neither EM-DAT nor GDIS can cover

all disaster details and impacts comprehensively. Additionally, EM-DAT data sometimes lacks exact start and end dates for certain disasters, which might mislead the analysis. Hence, there is still room for improvement in understanding the socio-economic impacts of hazards using accurate datasets or on-ground insights. In this study, we developed and compared CDI with the main widely used indices; however, CDI should also be compared with other drought monitoring indices to establish its superiority. One of the other limitations of this research is not considering hydrological variables (ground water, surface

runoff etc.) in the development of CDI. Due to the lack of availability of high-resolution hydrological data responsible for droughts, hydrological variables have not been included in this study. This omission might have caused some disparities in detecting GDIS results. Moreover, there are other methods and techniques that could be used to compute weights in CDI, which should be explored further for a better understanding of droughts and their socio-economic impacts.

## 6. Conclusion

Droughts rank among the most dangerous natural disasters, influencing a wide array of factors both directly and indirectly, with significant impacts on various socio-economic sectors. Despite numerous techniques and indices for analyzing the physical characteristics of droughts, the methods for understanding their direct socio-economic impacts remain underexplored, particularly at subnational levels. Our study addressed this research gap by investigating the direct propagation of agro-climatological droughts to socio-economic impacts using the GDIS dataset, which provides socio-economic disaster

information. Although several indices exist for drought assessment, many rely on single input parameters and fail to consider a comprehensive range of contributing factors, limiting their effectiveness in representing the socio-economic impacts of droughts. To address this limitation, we developed a new Combined Drought Indicator (CDI) that integrates two agricultural and two meteorological parameters to assess drought conditions. This novel index surpassed the performance of four commonly used single-parameter-based traditional indices, demonstrating superior accuracy in identifying GDIS droughts and

effectively representing their socio-economic impacts. The takeaways from this study can be summarized as follows:



- Globally, the multiparameter approach of the CDI proves to be a highly useful tool for assessing agro-climatological droughts compared to commonly used single-parameter-based traditional indicators such as SPI, SSI, NDVI, and STI.
- CDI-derived drought clusters exhibit a statistically significant representation of GDIS drought events (indicative of socio-economic impacts), with 95% of the GDIS events successfully identified using the CDI.
- The GDIS dataset provides direct socio-economic impact information and can be a valuable validation tool for drought indices.
- Regions characterized by the highest frequencies of drought events, as identified by GDIS and multiple indices, including CDI, are predominantly located in Sub-Saharan Africa, South Asia, South America, and Central America, underscoring the heightened vulnerability of these areas to drought occurrences.

This study underscores the applicability of CDI in analyzing droughts with enhanced precision compared to individual indices, effectively capturing their socio-economic repercussions. The direct usability of this technique worldwide could advance drought monitoring systems and inform policy development aimed at addressing both the socio-economic and agro-climatological impacts of droughts.

**Data availability**

All data and codes are available from the corresponding authors upon request.

**Author contributions**

SK and YS designed the study, with SK executing the work. SK prepared the manuscript with contributions from YS, YB, and BW. YS secured the funding.

**Competing interests**

The authors declare that they have no conflict of interest.

**Acknowledgments**

We gratefully acknowledge the financial support provided by the Japan Aerospace Exploration Agency (grant nos. ER3AMF106), the Japan Society for the Promotion of Science (grant no. 21H01430), and the Katsu Kimura Research Award for enabling this research.

**Financial support**

Japan Aerospace Exploration Agency (grant nos. ER3AMF106), the Japan Society for the Promotion of Science (grant no. 21H01430), and the Katsu Kimura Research Award.





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





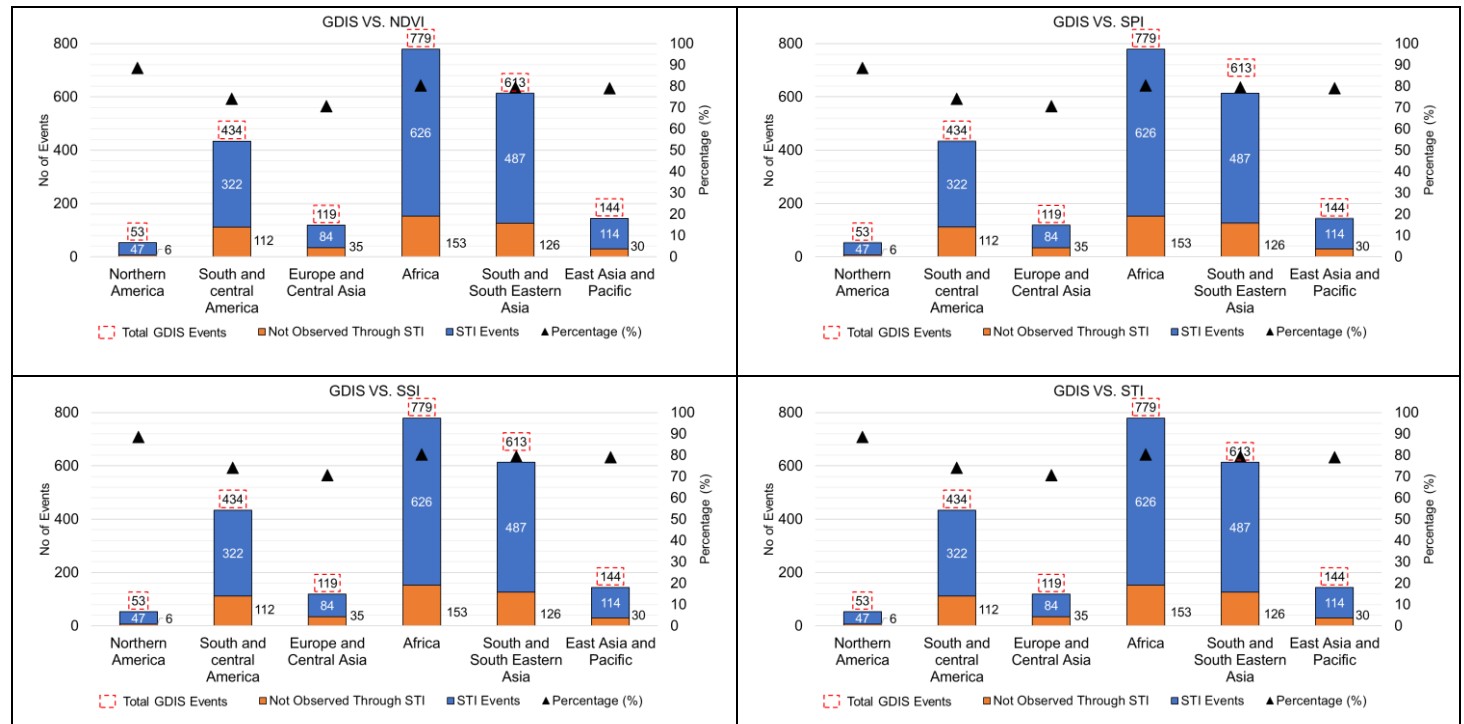

Appendix 2: Performance of Traditional Indices in Detecting GDIS Events Across Different Geographic Regions.

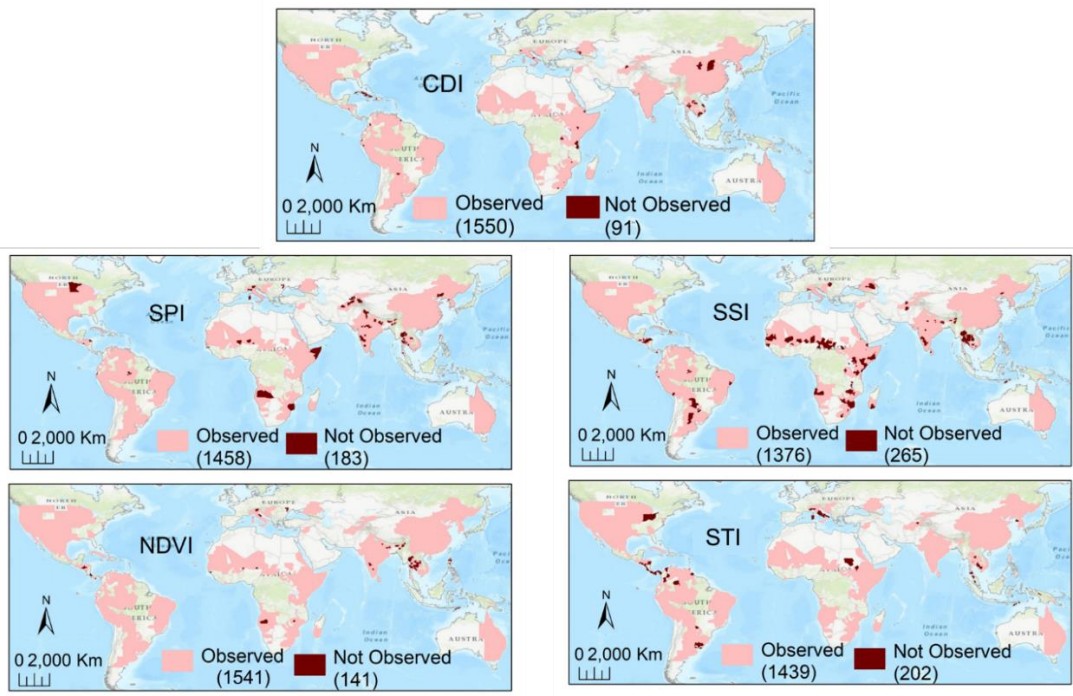

Appendix 3: Spatial locations wise performance of CDI, SPI, SSI, NDVI, and STI in detecting GDIS events, where





consistent events detected by each index are shown in pink, while inconsistent GDIS events (not observed) are shown in dark brown. The drought identification criteria were set to events during the actual drought period without considering short droughts (total GDIS events count = 1641).

865        Appendix 4: Performance of Traditional Drought Indices in Detecting GDIS Events Using Multiple Criteria

| SPI vs GDIS | Total Event | Drought Criteria | Actual Event Period (AEP) | | One Month Prior + AEP | | Two Month's Prior + AEP | | Three Month's Prior + AEP | |
|---|---|---|---|---|---|---|---|---|---|---|
| | | | -1 | 0 | -1 | 0 | -1 | 0 | -1 | 0 |
| Including Short Drought | 2142 | Observed | 1740 | 2114 | 1886 | 2140 | 1898 | 2142 | 1973 | 2142 |
| | | Not Observed | 402 | 28 | 256 | 2 | 206 | 0 | 160 | 0 |
| No Short Drought (event >=2 months) | 1641 | Observed | 1458 | 1641 | 1522 | 1641 | 1550 | 1641 | 1586 | 1641 |
| | | Not Observed | 183 | 0 | 119 | 0 | 91 | 0 | 82 | 0 |

| SSI vs GDIS | Total Event | Drought Criteria | Actual Event Period (AEP) | | One Month Prior + AEP | | Two Month's Prior + AEP | | Three Month's Prior + AEP | |
|---|---|---|---|---|---|---|---|---|---|---|
| | | | -1 | 0 | -1 | 0 | -1 | 0 | -1 | 0 |
| Including Short Drought | 2142 | Observed | 1770 | 2106 | 1912 | 2130 | 1936 | 2140 | 1988 | 2142 |
| | | Not Observed | 372 | 36 | 230 | 12 | 206 | 2 | 154 | 0 |
| No Short Drought (event >=2 months) | 1641 | Observed | 1376 | 1632 | 1572 | 1640 | 1580 | 1641 | 1610 | 1641 |
| | | Not Observed | 265 | 9 | 69 | 1 | 61 | 0 | 31 | 0 |

| NDVI vs GDIS | Total Event | Drought Criteria | Actual Event Period (AEP) | | One Month Prior + AEP | | Two Month's Prior + AEP | | Three Month's Prior + AEP | |
|---|---|---|---|---|---|---|---|---|---|---|
| | | | -1 | 0 | -1 | 0 | -1 | 0 | -1 | 0 |
| Including Short Drought | 2142 | Observed | 1867 | 2104 | 1983 | 2116 | 2028 | 2125 | 2003 | 2142 |
| | | Not Observed | 205 | 38 | 159 | 26 | 114 | 17 | 100 | 0 |
| No Short Drought (event >=2 months) | 1641 | Observed | 1541 | 1622 | 1561 | 1626 | 1574 | 1627 | 1576 | 1641 |
| | | Not Observed | 100 | 19 | 80 | 15 | 67 | 14 | 52 | 0 |





| STI vs GDIS | Total Event | Drought Criteria | Actual Event Period (AEP) | | One Month Prior + AEP | | Two Month's Prior + AEP | | Three Month's Prior + AEP | |
|---|---|---|---|---|---|---|---|---|---|---|
| | | | -1 | 0 | -1 | 0 | -1 | 0 | -1 | 0 |
| Including Short Drought | 2142 | Observed | 1680 | 2114 | 1740 | 2136 | 1820 | 2138 | 1810 | 2140 |
| | | | | | | | | | | |
| | | Not Observed | 462 | 28 | 292 | 6 | 272 | 4 | 131 | 2 |
| | | | | | | | | | | |
| No Short Drought (event >=2 months) | 1641 | Observed | 1439 | 1619 | 1479 | 1628 | 1510 | 1654 | 1556 | 1603 |
| | | | | | | | | | | |
| | | Not Observed | 202 | 22 | 162 | 13 | 131 | 7 | 85 | 0 |
| | | | | | | | | | | |
