# Peer review of "Global Assessment of Socio-Economic Impacts of Subnational Droughts: A Comparative Analysis of Combined Versus Single Drought Indicators"

_Hydrology and Earth System Sciences, 2024_

## Referee Comment (RC2)

**Journal: Hydrology and Earth System Sciences**

Title: Global Assessment of Socio-Economic Impacts of Subnational Droughts: A Comparative Analysis of Combined Versus Single Drought Indicators

https://doi.org/10.5194/hess-2024-245

**General comments**

This study analyses the propagation of drought hazards to socio-economic impacts using GDIS data, incorporating multiple drought indices and developing a novel CDI. The results indicate that CDI outperforms other indices, underscoring its utility in risk assessment and prioritisation of affected areas. The scope of the study fits well with the journal's theme of the study interactions with human activity, particularly in relation to droughts1.

Although the paper is well-written and structured, it does present some limitations in addressing the socio-economic aspects under study. Additionally, the results provided are insufficient to definitively support the conclusions. Based on the results presented in this paper, it cannot be definitively stated that the CDI index alone can determine the existence of socio-economic drought. After reviewing the manuscript and based on these comments, I recommend that the manuscript be reconsidered after a **major revision** to address the identified shortcomings.

**Specific comments**

The work uses the Global Disaster (GDIS) dataset distributed by SEDAC NASA to identify study areas, which are referred to as GDIS drought events. It is considered that the socio-economic variables used by this database to classify the area as affected should be detailed more precisely, since, as indicated in the discussion, vulnerability depends on the degree of development of the country in which it is located, and therefore the characterisation of the socio-economic variables is an important aspect to consider. The introduction should be expanded to include a detailed description of the socio-economic aspects related to drought and present the state of the art in this field.

This study exclusively utilises climatic variables, such as rainfall and temperature, along with indices like soil moisture and the vegetation index NDVI, without incorporating socio-economic variables. It is important to explain why these variables are not included in characterising drought events.

Given the global scope of this work, only climatological and vegetation predictors are utilised, without considering any socio-economic factors. An important question arises: Are there regions where the proposed index identifies periods of drought that are not recorded as such by the GDIS database? Figure 5 illustrates that certain areas experiencing extreme drought are not within any GDIS polygons. Additionally, it is noteworthy that Figure 5 does not include any European countries.

In section 2.3 it is indicated that soil moisture is obtained using a weighting for each of the strata; however, it is not detailed how it is obtained, and the way in which this is done should be explained, since it is one of the variables used to characterise the combined drought index, and as shown in the weights in Figure 4, this variable is quite important in the determination of this combined factor.

Section 2.5 states that '*we assumed January as the starting month and December as the end month of the respective event, and further analysis was carried out*'. Taking into account that the hydrological year in many databases is from October to September, it

would be important to know what percentage of the data used assume an unknown period, and to analyse the sensitivity of the results obtained to a possible alteration of the hydrological year.

Regarding the results section, the results are presented in absolute terms by quantifying the number of detected and undetected events, which makes them difficult to understand. The quantification of the accuracy of the proposed methodology should be done using specific metrics obtained from a confusion matrix, detailing: Accuracy, Precision, recall, specificity, F1-score, AUC, etc. This will allow the discussion section to be completed by comparing similar metrics from previous work.

From the visual analysis of Figure 7, it cannot be concluded that the combined index is clearly better than the individual indices, as indicated in line 525 as the results represented in that figure the combined index performs almost as well as the NDVI and in some cases slightly worse.

The analysis of the accuracy of the different indices could be completed with information on the socio-economic characteristics of each region as well as the typology of land cover in the region analysed, the accuracy metrics according to these variables in order to be able to conclude in a quantifiable way under which conditions one index performs better than another.

From line 470 to Figure 9, the purpose of testing the correlation between the combined index and the indices should be clarified. Considering that the combined index has been obtained from a principal component analysis of these variables, it is logical that it is correlated with the different parameters according to the weighting weights, it should be clarified what the purpose of this analysis is.

As mentioned above, there are conclusions that are not justified by the results presented. Lines 593 to 595, The following is stated: '*This novel index surpassed the performance of four commonly used single-parameter-based traditional indices, demonstrating superior accuracy in identifying GDIS droughts and effectively representing their socio-economic impacts*'. However, as mentioned above, this conclusion is not supported by the information and results presented in this work.

Lines 599 to 600: The following is stated: *'CDI-derived drought clusters exhibit a statistically significant representation of GDIS drought events (indicative of socio-economic impacts), with 95% of the GDIS events successfully 600 identified using the CDI'*. However, to support this assertion, it is necessary to consider the full set of metrics from the confusion matrix. While the index may identify GDIS-catalogued drought events, it could also detect other drought events that do not necessarily have socio-economic effects. Based on the results presented in this paper, it cannot be conclusively stated that the CDI index alone can determine the existence of socio-economic drought.

**Technical corrections**

It should be clarified whether the ranges in Figure 3 correspond to the classification set out in Figure 1, and if so, the nomenclature should be homogenised.

Figure 7 has very low resolution, however, as it is proposed to replace the figure with a display of the results in terms of the metrics of the confusion matrix, I understand that this figure will be replaced in its new format.

---

## Author Comment (AC1)

Response to Reviewer 1 Comments

MS No.: hess-2024-245

Thank you, Dr. Jasmin Heilemann, for your thoughtful review of our manuscript. We sincerely appreciate the time and effort you dedicated to carefully evaluating our work and providing valuable comments and suggestions. Your insights have significantly contributed to improving the quality of our manuscript. We have addressed each of your comments accordingly, and our responses are provided below. Reviewer comments are highlighted in red, while our responses are in black.

**General comments:**

In general, it is an interesting paper within the scope of the journal that addresses the highly relevant issue of detecting socio-economic impacts of drought using biophysical indicators on a global scale. The paper proposes a Combined Drought Indicator (CDI), which is constructed by using single-input based drought indices (SPI, NDVI, SSI, STI) and performing a Principal Component Analysis (PCA). The authors show that the CDI outperforms the single-input drought indices in its ability to capture drought events observed in the global GDIS dataset.

While the paper presents a very relevant analysis with noteworthy results, what is currently missing from the paper is a discussion of the benefits of using the PCA method to construct the CDI. This discussion should include details of the benefits of PCA for the CDI, as well as a discussion of the applicability of the CDI for (regionalized) drought impact monitoring and prediction. Including this in the manuscript would significantly enhance the paper and give greater significance to the implications, with potential applications of the CDI beyond this paper. My specific comments are listed below.

**Specific comments:**

1. Title: Global analysis of sub-national droughts: "Sub-national" droughts sound misleading, as droughts are not constrained by national borders. If possible, rephrase as "droughts at sub-national scale", or similar.

> ➢ Thank you for recommending a title change for the manuscript. We agree that the original title may have been somewhat misleading in reflecting the study's scope. As per your suggestion, we have revised the title to more accurately align with the analysis and findings presented. The new title is: "Global Assessment of Socio-Economic Drought Events at Sub-National Scale: A Comparative Analysis of Combined Versus Single Drought Indicators."

2. Abstract: "Out of 2142 drought events in 2001-2021 recorded by GDIS, NDVI, SSI, SPI, and STI identified 1867, 1770, 1740, and 1680 drought events, respectively. […] CDI outperformed the other single-input-based drought indices and identified 1885 events." Consider adding percentages or otherwise present convincing quantitative results that show the superiority of the CDI more directly.

Thank you for pointing this out. We agree that adding percentages alongside event counts will be helpful for readers to better understand the results. In the revised manuscript, the percentages have been included.

The revised sentence in the abstract is as follows, "Out of 2142 drought events in 2001-2021 recorded by GDIS, NDVI, SSMI, SPI, and STI identified 1867 (87.16%), 1770 (82.63%), 1740 (81.23%), and 1680 (78.43%) drought events, respectively."

3. Fig. 1: It is a bit confusing to frame the "wet" conditions as drought categories. Would it be possible to find a different notion, e.g. moisture? Or Drought/Wetness category?

Thank you for your comment. We understand the concern regarding the terminology. However, the drought classes and categories used in our study are based on the Standardized Precipitation Index (SPI) (McKee et al.,1993), which is a widely accepted approach. This classification system, which includes both wet and dry conditions under a common drought/wetness scale, has been adopted by several major drought monitoring institutions, such as the National Drought Mitigation Center (USA) and the Australian Bureau of Meteorology. Accordingly, we have followed these established references.

A clarifying sentence has been included in the revised manuscript (Section 3.1, line 210-211), which reads: "The drought categories applied in this study follow the widely used SPI classification, originally developed by McKee et al. (1993)."

4. Lines 116-125: The topic of the paper are socio-economic impacts of droughts. However, socioeconomic impacts manifest very differently in different sectors. E.g., in the introduction, you mention urban areas/water shortages in dams (lines 25-36). The drought indices you chose (SPI, NDVI, SSI, STI) are mostly useful for the ag sector. Please elaborate on how this affects the results, and if/how the CDI can capture socio-economic impacts isen non-ag sectors, e.g. urban areas.

Thank you for this thoughtful. We agree that the socio-economic drought impacts vary significantly across sectors and that the indices used in our study, SPI, NDVI, SSMI, and STI, are more closely aligned with impacts on the agricultural and vegetation-related sectors.

To address this, we have revised the manuscript to better explain the broader relevance of CDI beyond the agricultural context. Specifically, we now emphasize that the CDI integrates multiple environmental variables (SPI, NDVI, SSMI, and STI), which collectively capture drought conditions with potential implications for non-agricultural sectors as well. For example, SPI and SSI are closely related to hydrological drought, which can impact urban water supply and reservoir levels, while STI may relate to energy demand, health, and heat stress in urban settings (Vicente-Serrano et al., 2012; Hao & Singh, 2015; Wang et al., 2020).

We have also added a discussion on future research directions, highlighting the need for sector-specific indicators and data (e.g., urban water demand, infrastructure vulnerability, or energy production) to enhance the socio-economic relevance of CDI. These changes have been added in the revised manuscript in the Discussion section (Line 625-630), as follows:

"The current CDI primarily reflects agro-environmental droughts due to the nature of its input indices. However, since it combines precipitation, temperature, soil moisture, and vegetation data, it may also capture broader drought signals relevant to urban systems, such as water availability and heat stress (Vicente-Serrano et al., 2012; Hao & Singh, 2015; Wang et al., 2020). In future work, we aim to enhance CDI by incorporating sector-specific indicators to better assess socio-economic impacts beyond agriculture."

5. Section 3.2: Here, I miss a description of the reasons why the PCA method was chosen to construct the PCA. This is the main innovation of the paper, and should therefore be featured more prominently, also in the introduction. E.g., explain what the added value of the PCA is compared to other techniques to compute a CDI. Why are regression-based approaches not used? (e.g. is it an advantage that the PCA does not have a dependent variable?)

Thank you for this insightful comment. We agree that the reasoning for using PCA to construct the CDI should be made clearer and more prominent, particularly given its central role in our methodology. Accordingly, we have revised the Methods sections to elaborate on this point. The primary reason for selecting PCA is its ability to reduce dimensionality while preserving the maximum possible variance from the original data. Unlike regression-based approaches, PCA does not require a dependent variable, which is particularly advantageous in our context where the aim

is to integrate multiple independent agro-climatological indicators into a single composite metric without presupposing a specific impact model.

Furthermore, PCA generates orthogonal (i.e., uncorrelated) components, allowing us to avoid multicollinearity issues that often arise in regression models. This ensures that each input variable contributes uniquely to the CDI. The resulting weights derived from PCA reflect the actual variability and importance of each index within the combined space, making the CDI more representative of spatio-temporal drought conditions across diverse climates. While regression methods could be used if a target impact (e.g., crop yield, water stress) were clearly defined and globally available, our goal was to develop a generalized, impact-sensitive drought indicator suitable for global application and validation against GDIS events.

To address your suggestion, we have now clarified these points in the revised manuscript and added the following text to Section 3.2, line 230-239:

"The PCA method was selected for constructing the CDI due to its ability to extract dominant patterns of variability across multiple input indices without requiring a dependent variable. This makes it particularly suitable for integrative assessments across diverse drought types and geographic regions. Compared to other commonly used weighting methods, such as the Analytic Hierarchy Process (AHP), which relies on expert judgment (Saaty, 1980), or entropy weighting, which uses the diversity of information in data (Mahato et al., 2023), PCA offers an objective, data-driven approach that reduces subjectivity. While regression-based methods have also been explored to link drought indicators with socio-economic impacts (Hao et al., 2014b), they typically require clearly defined response variables and may introduce model-based biases. In contrast, PCA generates uncorrelated components and assigns weights based on explained variance, enhancing reproducibility and generalizability in global-scale assessments."

6. Lines 227-229: What is the total number of observations for the single-based drought indices used in the PCA? Does this number meet the requirements of the no. of observations usually applied in PCA? Please specify.

Thank you for your valuable comment. In our study, PCA was performed separately for each calendar month, using time series data from 21 years (2001–2021). As a result, each monthly PCA was based on 21 observations per variable (SPI, STI, SSMI, and NDVI), giving a subject-to-variable ratio of 5.25:1.

While this sample size is relatively small, it is within the acceptable range for PCA applications, especially given the low number of input variables. According to established guidelines, a minimum subject-to-variable ratio of 5:1 is considered acceptable for stable PCA solutions when the dataset is not highly noisy (Jolliffe, 2002; Gorsuch, 1983). Moreover, the purpose of PCA in our study is to derive objective, data-driven weights rather than to interpret component loadings across multiple dimensions. This limited yet structured approach enabled us to compute monthly-specific weights that reflect seasonal variability in the relationship between drought indicators.

We have revised the manuscript and added an explanation to justify the suitability of the sample size (Section 3.2, Lines 240 - 247).

"In this study, the PCA technique was used to assign monthly weights to the four input indices: SPI, STI, SSMI, and NDVI. PCA is commonly used in environmental and climate studies to extract dominant patterns in multivariate datasets without requiring a dependent variable. In this study, PCA was conducted separately for each calendar month using time series data from 2001 to 2021, resulting in 21 observations per variable. Although the number of observations is relatively modest, it satisfies the commonly accepted subject-to-variable ratio of at least 5:1 for PCA (Jolliffe, 2002; Gorsuch, 1983), especially when the number of variables is low, and the objective is dimensionality reduction. Through PCA, new orthogonal (independent of each other) variables, i.e., P.C.s, were constructed using linear combinations of the original indices without significant loss of information."

7. Lines 258: "…has been widely accepted in previous work." Which previous work? Please provide citations. Please extend this to the other text when you mention previous work without giving references.

Thank you for pointing this out. We agree that such text must include proper referencing. Somehow, this was previously overlooked. In the revised manuscript, all such sentences referring to previous work are now properly cited with the appropriate references. The revised manuscript includes the following modifications at suggested place (section 3.3 line 280);

"Figure 1 illustrates that values below -1 indicate moderately to extremely dry conditions, which have been widely accepted in previous works (McKee et al., 1993; Bayissa et al., 2019a; Kulkarni et al., 2020b)." (section 3.3, line 280)

McKee, T. B., Doesken, N. J., and Kleist, J.: The relationship of drought frequency and duration to time scales, in: Proceedings of the 8th Conference on Applied Climatology, 17–22, Am. Meteorol. Soc., Anaheim, CA, 1993.

Bayissa, Y. A., Tadesse, T., Svoboda, M., Wardlow, B., Poulsen, C., Swigart, J., and Van Andel, S. J.: Developing a satellite-based combined drought indicator to monitor agricultural drought: a case study for Ethiopia, GISci Remote Sens, 56, 718–748, https://doi.org/10.1080/15481603.2018.1552508, 2019b.

Kulkarni, S. S., Wardlow, B. D., Bayissa, Y. A., Tadesse, T., Svoboda, M. D., and Gedam, S. S.: Developing a remote sensing-based combined drought indicator approach for agricultural drought monitoring over Marathwada, India, *Remote Sens.*, 12, 2091, https://doi.org/10.3390/rs12132091, 2020b.

Similarly, references have been added to line no 575 as follows:

Previous studies, whether regional or global, often relied on single-parameter-based indices for drought monitoring, such as SPI (McKee et al., 1993), NDVI (Ji & Peters, 2003), or soil moisture indices (Liu et al., 2012).

McKee, T. B., Doesken, N. J., and Kleist, J.: The relationship of drought frequency and duration to time scales, Proc. 8th Conf. on Applied Climatology, Anaheim, CA, USA, 17–22 January 1993, American Meteorological Society, Boston, MA, 179–183, 1993.

Ji, L. and Peters, A. J.: Assessing vegetation response to drought in the northern Great Plains using vegetation and drought indices, Remote Sens. Environ., 87, 85–98, https://doi.org/10.1016/S0034-4257(03)00174-3, 2003.

Liu, Y. Y., Parinussa, R. M., Dorigo, W. A., de Jeu, R. A. M., Wagner, W., van Dijk, A. I. J. M., McCabe, M. F., and Evans, J. P.: Developing an improved soil moisture dataset by blending passive and active microwave satellite-based retrievals, Hydrol. Earth Syst. Sci., 15, 425–436, https://doi.org/10.5194/hess-15-425-2011, 2011.

8. Lines 258-265: The thresholds of the drought indices used for detecting the drought impacts listed in the GDIS dataset are very crucial, though the explanation remains too vague (it's a simple process, but I had to read over the section several times to understand this). Please make this process more explicit, e.g. via adding a table. Also, I miss a clear explanation of how the spatial scales between the gridded drought indices and the sub-national GDIS events are matched for the detection of drought impacts (is it counted as drought event if more than half of the pixels in the GDIS area show a deviation below the drought threshold? Or do you first calculate the average of the drought indices across all grid points and then compare it with the thresholds?)

We thank the reviewer for this valuable comment. We agree that greater clarity was needed regarding both the thresholding methodology and the spatial aggregation procedure.

In the revised manuscript, we have now substantially expanded the explanation in this section to clearly describe the consistency assessment process. Specifically:

1. Threshold criteria: For each GDIS drought event (defined by a spatial polygon and temporal range), we extracted the corresponding gridded monthly values of five drought indices: SPI, STI, SSMI, NDVI, and CDI. A given index was considered consistent with the GDIS event if any month within the event's timeframe had a spatially averaged index value within the polygon that was below a set threshold. We used two thresholds in our analysis. A primary threshold of -1, which corresponds to moderate to extreme drought conditions and an alternative threshold of 0, used for sensitivity analysis.

2. Spatial aggregation: To answer the spatial scale difference between gridded indices and the polygonal GDIS data, we computed the mean value of each drought index across all grid cells located within the GDIS polygon for each month. This monthly mean was then compared to the

drought threshold. This approach was chosen over a pixel-count method to ensure consistency and simplicity across different event sizes and spatial resolutions.

Yes, the assessment was based on the average index value across all grid cells within each polygon, which was then compared to the threshold, rather than using a pixel-counting or majority-area approach.

Clarification via a table: We have added table to the appendix (appendix 1), of revised manuscript to summarize the step wise process used in evaluating index consistency with GDIS drought events. This should make the methodology more accessible and easier to follow.

Appendix 1: Step wise procedure for assessing the consistency of gridded drought indices with GDIS drought events

| Step | Description |
| --- | --- |
| 1 | For each GDIS event, extract spatial polygon and event time range (referring EM-DAT for event details) |
| 2 | Extract monthly gridded index values (SPI, STI, SSI, NDVI, CDI) within polygon |
| 3 | Compute monthly spatial average of index values within polygon |
| 4 | Check if any month in the event has an average value below threshold |
| 5 | If yes, mark that index as consistent with the GDIS event |
|  |  |

This revised writeup can be found in section 3.3, from line 269 to line 289 in the revised manuscript. The revised version is as follows.

"The GDIS polygons were overlaid onto the gridded drought index (SPI, STI, SSMI, NDVI, and CDI) layers separately, to extract spatial and temporal raster-based information for each drought event. A total of 2,142 events recorded between 2001 and 2021 were analyzed. Event-specific details such as location and start and end dates were obtained from the GDIS and EM-DAT databases. For each drought event, index values were extracted from the respective raster datasets based on the spatial extent (polygon) and duration of the event. For example, if a GDIS event occurred in Bihar, India, from March to December 2012, the relevant monthly raster values for SPI, STI, SSMI, NDVI, and CDI within the Bihar polygon during that period were extracted.

To align the gridded drought indices with the spatial scale of the GDIS events, we computed the monthly spatial average of each drought index over all grid cells within the corresponding event polygon. This process produced a single time series per index for each event. The resulting geodatabase tables were then analyzed to assess whether the index values were consistent with the

GDIS records. Following previous literature (McKee et al., 1993; Bayissa et al., 2019a; Kulkarni et al., 2020b), a threshold of ≤ -1 was used to define moderate to extreme drought conditions (Figure 1). A drought index was considered consistent with a GDIS event if the average value of the index within the event polygon was ≤ -1 in any month during the event's duration. For instance, if the average STI value within the Bihar polygon fell below -1 in any month between March and December 2012, STI would be considered consistent with that GDIS event. To evaluate sensitivity, a secondary analysis was also conducted using a threshold of < 0. In this case, if any monthly average value of an index was below zero during the event, it was also marked as consistent. This two-threshold approach allowed for both conservative and more inclusive assessments of drought index performance against GDIS events. For clarity, a step wise procedure of thresholding and spatial averaging is presented in appendix 1."

9. Line 327: Please specify why you chose April as the month for displaying the PCA results. Does it represent the yearly average best? How important are intra-annual fluctuations? April is not a typical drought month in the northern or southern hemisphere.

We thank the reviewer for this observation. The PCA analysis was conducted for all twelve months to capture the seasonal variation in drought patterns across different indices. However, due to space constraints in the manuscript, we were unable to include the full set of monthly results.

April was selected as a representative example to present in the manuscript as it falls between typical dry and wet seasons in many parts of the world and hence does not overly bias the patterns toward either extreme. While April is not necessarily a peak drought month in either hemisphere, it offers a mid-season picture that helps demonstrate the spatial structure of the PCA components without being overly influenced by strong seasonal extremes.

We acknowledge the importance of intra-annual fluctuations and confirm that similar analyses were carried out for each month. The weights computed using PCA for each month and each input variable are provided in the supplementary information of the revised manuscript (Appendix 2). The resulting maps and general interpretation are as follows;

The monthly PCA-derived weight maps show clear seasonal and spatial variation in the importance of each drought input variable. Rainfall (SPI) generally carries higher weights in monsoon-dependent and rain-fed regions during their respective wet seasons, especially in South Asia and sub-Saharan Africa. Soil moisture (SSMI) shows consistently moderate to high weights across temperate and tropical agricultural zones, reflecting its relevance for root-zone drought. NDVI contributes more in heavily vegetated regions like the Amazon, Central Africa, and Southeast Asia during growing seasons, with reduced influence in arid and non-vegetated areas. LST (via TCI) has moderate weights in regions prone to heat stress, particularly during summer in the Northern Hemisphere. Together, these dynamic weights demonstrate CDI's adaptability to seasonal climatic conditions and regional drought sensitivities.

[Figure]

(a)

Weights (Rainfall)

| | | |
|---|---|---|
| ■ 0 - 0.1 | ■ 0.2 - 0.3 | ■ 0.4 - 0.5 |
| ■ 0.1 - 0.2 | ■ 0.3 - 0.4 | ■ 0.5 - 0.6 |

■ 0.6 - 0.7

N

0    12,500    25,000 Km

[Figure]

Weights (LST)

(b)

0 - 0.1   0.2 - 0.3   0.4 - 0.5   0.6 - 0.7
0.1 - 0.2   0.3 - 0.4   0.5 - 0.6

0   12,500   25,000 Km

Weights (NDVI)

(c)

0 - 0.1   0.2 - 0.3   0.4 - 0.5   0.6 - 0.7
0.1 - 0.2   0.3 - 0.4   0.5 - 0.6

0   12,500   25,000 Km

[Figure]

Appendix 2. Monthly spatial distribution of PCA-derived weights for rainfall (a), temperature (b), NDVI (c), and soil moisture (d) used in CDI computation. Color scales indicate the relative contribution of each variable to CDI (brown = low weight, green = moderate weight, blue = high weight).

10. Table 2: You show the false-negative (when a GDIS drought event existed, but the drought index did not indicate a drought event) in the table as "not observed". Likewise, what is the rate of false-positive cases (how often did the drought index indicate a drought event not reported in the GDIS?)? You discuss this in the text (lines 390ff), but it would be beneficial for the reader to understand the magnitude of these cases in numbers.

We thank the reviewer for this valuable observation. While Table 2 presents false-negative cases, where GDIS-reported drought events were not detected by the indices, we agree that understanding the magnitude of false-positive cases/instances where an index (such as CDI) indicated drought conditions in locations or periods not reported in GDIS, is equally important.

As mentioned in the manuscript (lines 435), we did observe several such cases visually. For example, the CDI detected drought conditions in South Argentina (2014–15), Namibia (2013), and

parts of Europe (2018), which were not captured in GDIS. However, we did not perform a comprehensive quantitative analysis of false positives, primarily due to two key limitations:

1. Incomplete event coverage in GDIS/EM-DAT: Though GDIS gives drought event information, it does not comprehensively cover all real-world drought events, particularly in developing regions with limited reporting infrastructure or where impacts may not meet the threshold for international disaster recording. As a result, a drought detected by an index but missing from GDIS may not necessarily represent a false positive, but rather a real event that went undocumented. This reporting bias makes it challenging to confidently interpret such mismatches as false positives.

2. Lack of defined temporal frames for reverse analysis: Unlike GDIS, which provides explicit event start and end dates, drought indices can show anomalies across various timeframes (monthly, seasonal, annual), making it difficult to define a standard "event" period in the absence of an external reference. Applying a consistent and unbiased reverse framework for identifying false positives is, therefore, not straightforward and risks misclassification.

Given these limitations, we restricted our analysis to a visual identification of potential false-positive instances. However, we fully agree that a quantitative false-positive assessment would be a valuable future direction. With access to more detailed, high-resolution, and timely impact datasets, particularly in underrepresented regions, this could be systematically explored.

We have included this discussion in the revised manuscript (Discussion section, Line 659) as follows:

"While this study focused on false-negative cases using GDIS as a reference, a systematic assessment of false-positive cases remains challenging. This is due not only to the lack of defined temporal frames for reverse analysis but also to the incomplete coverage of drought impacts in GDIS, especially in developing regions where many drought events may go unreported. These limitations could be addressed in future research using more comprehensive and high-resolution impact datasets."

11. Discussion: In the discussion, an important point would be how the CDI could be used/applied for drought impact forecasting and/or policy-making. Could the CDI (computed via PCA) help to improve drought impact forecasting? How does the regionalization of the CDI affect the capacity to be used for that purpose?

Thank you for this insightful comment. We agree that discussing the practical applications of CDI, particularly for drought impact forecasting and policy-making, is important. We have now included a brief discussion in the revised manuscript (Section 5: Discussion, lines 654 - 659) addressing this point. The added text is as follows;

"By integrating multiple indicators, CDI provides a more comprehensive view of drought conditions that is useful for identifying at-risk areas. For example, in regions like East Africa or

Central India, where both rainfall deficits and vegetation stress are common during droughts, CDI captures these multiple dimensions more effectively than single-parameter indices. Its regionalized structure ensures better alignment with local climate dynamics, enhancing its potential utility in forecasting and policy targeting. With adaptation to near-real-time inputs, the CDI framework could support early warning systems and guide proactive measures such as crop insurance triggers or water allocation planning."

12. Lines 570ff: You could additionally mention that text mining is a research field potentially providing alternative impact databases for droughts next to the GDIS.

We thank the reviewer for this valuable suggestion. We agree that text mining and natural language processing (NLP) approaches are emerging as promising tools for generating alternative or supplementary drought impact datasets. These methods have the potential to fill gaps in existing databases such as GDIS and EM-DAT by extracting event-specific impact information from news reports, social media, and institutional records. We have now acknowledged this point in the revised discussion section at lines 679-683, as follows:

"Emerging approaches such as text mining and natural language processing (NLP) offer promising pathways to address this gap by automatically extracting drought impact information from news articles, institutional reports, and social media (Fritz et al., 2019; Sathianarayanan et al., 2024), and could serve as alternative or supplementary impact datasets alongside GDIS and EM-DAT."

13. Line 550: "despite experiencing higher climatic anomalies, developed nations are less likely to be socio-economically affected …". This statement needs to be specified. It needs to become clear that the higher climatic anomalies relate to the local climate, and are not compared in absolute terms. A small anomaly in an already dry climate can provoke much more negative drought impacts compared to a larger anomaly in a wetter climate. Otherwise, this suggests that climate/drought impacts in developed nations are higher than in developing countries, which is not the case.

Thank you for pointing out this. We agree that the original phrasing may have been misleading. Our intention was not to suggest that developed nations experience greater absolute drought impacts, but rather that they may face significant climatic anomalies relative to their own baseline (e.g., unusually dry years), yet are often less socio-economically affected due to higher adaptive capacity and resilience. We have rewritten the sentence in the revised manuscript (Section-discussion, lines 641-643) to clarify our point as follows.

"Although developed nations may experience significant climate anomalies relative to their local climatic norms, they are generally less socio-economically impacted by droughts than developing countries, which tend to be more vulnerable due to limited adaptive capacity."

14. Lines 583-584: "Moreover, there are other methods and techniques that could be used to compute weights in CDI …" Like which methods? Please specify and give a short reason why they could be apt.

We thank the reviewer for this helpful comment. We have revised the manuscript to include specific alternative methods that could be used to compute weights in the CDI. These include entropy weighting, the analytic hierarchy process (AHP), and machine learning-based approaches such as random forest feature importance. Each method offers unique advantages: entropy weighting emphasizes data variability, AHP incorporates expert judgment, and machine learning techniques can capture nonlinear relationships between indicators and observed impacts. These details have been added to the discussion section of the revised manuscript at lines 690-694, as follows:

"Further, other alternative methods such as entropy weighting, the analytic hierarchy process, or machine learning-based feature importance (like random forests) could also be explored to compute weights in CDI, as they may better capture indicator relevance by incorporating data variability, expert knowledge, or nonlinear relationships with observed impacts.

"

**Technical corrections:**

Line 121: "combine drought indicator" -> correct to "combined"

Thank you. The word 'combine' has been corrected to 'combined' in the revised manuscript (Section 1: Introduction, line 121).

Thank you for pointing out this error. We somehow overlooked it. In the revised manuscript, a legend has been added to Figure 3a. The corrected figure is as follows:

[Figure]

Figure 3. Spatial Distribution of GDIS drought frequencies (a) Global scale (b) East Africa (c) America and (d) Asia. The drought frequencies range from one to eight, represented by shades ranging from light yellow to dark brown, respectively.

9: Second panel: "CDI vs. TCI", please correct to "CDI vs. STI"

Thank you for the correction. In the revised manuscript, CDI vs TCI has been changed to CDI vs STI. The revised figure is as follows,

[Figure]

**Figure 10. Spatial Correlation Between CDI and Single Input-Based Traditional Indices for a Sample Month (April): (a) CDI vs. SPI, (b) CDI vs. STI, (c) CDI vs. NDVI, and (d) CDI vs. SSMI. Negative correlations are represented in shades from yellow to red, while positive correlations are shown in shades from light green to dark green.**

Appendix 2: This figure shows four times the same plot. This should be corrected.

Thank you for pointing out this error. In the revised manuscript, the corrected images have been included. The corrected images are as follows,

[Figure]

Appendix 4: Performance of Traditional Drought Indices in Capturing GDIS Events Across Global Regions: Comparative Assessment of (a) NDVI, (b) SPI, (c) SSMI, and (d) STI

********************* Thank You *********************

---

## Author Comment (AC2)

**Response to Reviewer 2 Comments**

**MS No.: hess-2024-245**

Thanks to the reviewer for reviewing our manuscript. We sincerely appreciate the time and effort you dedicated to carefully evaluating our work and providing valuable comments and suggestions. Your insights have significantly contributed to improving the quality of our manuscript. We have addressed each of your comments accordingly, and our responses are provided below. Reviewer comments are highlighted in red, while our responses are in black.

**General comments**

This study analyses the propagation of drought hazards to socio-economic impacts using GDIS data, incorporating multiple drought indices and developing a novel CDI. The results indicate that CDI outperforms other indices, underscoring its utility in risk assessment and prioritization of affected areas. The scope of the study fits well with the journal's theme of the study interactions with human activity, particularly in relation to droughts1.

Although the paper is well-written and structured, it does present some limitations in addressing the socio-economic aspects under study. Additionally, the results provided are insufficient to definitively support the conclusions. Based on the results presented in this paper, it cannot be definitively stated that the CDI index alone can determine the existence of socio-economic drought. After reviewing the manuscript and based on these comments, I recommend that the manuscript be reconsidered after a **major revision** to address the identified shortcomings.

**Specific comments**

1. The work uses the Global Disaster (GDIS) dataset distributed by SEDAC NASA to identify study areas, which are referred to as GDIS drought events. It is considered that the socio-economic variables used by this database to classify the area as affected should be detailed more precisely, since, as indicated in the discussion, vulnerability depends on the degree of development of the country in which it is located, and therefore the characterisation of the socio-economic variables is an important aspect to consider. The introduction should be expanded to include a detailed description of the socio- economic aspects related to drought and present the state of the art in this field.

Thanks to the reviewer for this insightful comment. As suggested, we have expanded the introduction to provide a clearer overview of the socio-economic impacts of drought and the current state of the art in this field. We have edited the introduction section and new details are added a new paragraph (Lines 86 to 96) that discusses the variation in drought impacts based on development levels, the importance of socio-economic vulnerability, and the limitations of earlier impact datasets. We also explain how the recently developed GDIS dataset addresses these gaps by offering sub-national socio-economic impact data for global analysis.

Revised text (Lines 86 to 96):

"Droughts have significant socio-economic impacts, including crop losses, food insecurity, income reduction, water shortages, and displacement. The severity of these effects varies by region, depending on development level, infrastructure, and adaptive capacity. In high-income areas, systems like irrigation and insurance help reduce impacts, while in low-income regions, even moderate droughts can trigger crises (Brooks et al., 2005; and Pak-Uthai and Faysse, 2018). Recent studies (Panwar and Sen, 2020; and Udmale et al., 2014) highlight the importance of incorporating socio-economic vulnerability into drought assessments. However, the direct link between drought hazards and their socio-economic repercussions remains underexplored, partly due to the limited availability of reliable global impact data. Earlier efforts, such as the U.S. Drought Impact Reporter and the European Drought Impact Report Inventory, provided region-specific insights but lacked global coverage. To address this, the Geocoded Disaster (GDIS) dataset, developed from EM-DAT, offers geocoded disaster locations and detailed sub-national data on affected population, fatalities, and economic losses. By overcoming previous limitations, GDIS enables spatial analysis of drought impacts across diverse socio-economic contexts."

2. This study exclusively utilises climatic variables, such as rainfall and temperature, along with indices like soil moisture and the vegetation index NDVI, without incorporating socio-economic variables. It is important to explain why these variables are not included in characterising drought events.

Thank you for your comment. The main objective of our study is to evaluate the performance of combined versus single drought indicators by comparing them against observed drought event data from the GDIS database. GDIS provides subnational records of actual socio-economic drought-related events, and we use this dataset as an observational benchmark for this study.

In this study, we deliberately did not include socio-economic variables as part of the drought indicators. Our aim is to assess how well agro-climatological indices, such as precipitation, temperature, soil moisture, NDVI, and CDI, can reflect or correspond to real-world drought events rather than attempt to model those events directly. Including socio-economic variables in the indicators would have complicated the evaluation, as it would

mean comparing GDIS data against indicators that already include similar information, potentially biasing the results and reducing the objectivity of our assessment.

We acknowledge that some regional studies have incorporated socio-economic data; such as crop prices or agricultural losses, for drought monitoring (e.g., Brown & Funk, 2008; Lobell & Burke, 2010; Wang et al., 2022). However, these approaches are often limited to specific regions and lack the global consistency required for a study of this scale. Our scope is therefore distinct: rather than modeling socio-economic drought events, we evaluate the capacity of agro-climatological indicators alone to serve as reliable proxies for observed drought events globally.

Moreover, for operational drought monitoring and policymaking, particularly at global and subnational scales, consistent, high-resolution, and continuous datasets are essential. Currently available socio-economic datasets often lack such spatial and temporal consistency. Therefore, identifying agro-climatological indicators that closely align with observed drought events can help strengthen early warning systems and support more effective policy decisions, especially in data-scarce regions.

These details have also been included in the revised manuscript (Discussion section, lines 592 to 596) as follows:

"While some regional studies have used socio-economic data such as crop prices or agricultural losses for drought monitoring (e.g., Brown & Funk, 2008; Lobell & Burke, 2010), we exclude such variables to avoid overlapping with the GDIS, which already incorporates socio-economic factors like the number of affected individuals, deaths, and total damage caused by drought. Our focus is on evaluating how well agro-climatological indices, such as CDI, SPI, and NDVI, capture drought events based on climatic and environmental conditions. Additionally, globally consistent socio-economic datasets are often limited, and their availability may vary by region, making agro-climatological based indicators more practical and reliable for large-scale drought monitoring."

3. Given the global scope of this work, only climatological and vegetation predictors are utilised, without considering any socio-economic factors. An important question arises: Are there regions where the proposed index identifies periods of drought that are not recorded as such by the GDIS database? Figure 5 illustrates that certain areas experiencing extreme drought are not within any GDIS polygons. Additionally, it is noteworthy that Figure 5 does not include any European countries.

Thank you for this valuable observation. We fully agree with your point, and indeed, our analysis revealed several instances where the CDI detected drought conditions that were not recorded in the GDIS database. These discrepancies are acknowledged and discussed in detail in Section 4.2 (lines 432-440) of the manuscript as follows:

"CDI detected severe drought events in South Argentina during 2014–15, Namibia in 2013, and parts of Europe in 2018, which were not reflected in GDIS event records. These instances highlight that not all agro-climatic droughts lead to recorded socio-economic impacts, especially in regions with strong adaptation and mitigation capacities. Practices such as advanced irrigation, drought-resistant crop varieties, or effective early warning systems may help manage the agricultural and societal impacts of climatic stress, thereby reducing the likelihood of such events being recorded in GDIS. It is also important to note that GDIS does not comprehensively capture all real-world drought events, particularly in regions with limited reporting mechanisms or institutional capacity. As a result, some

drought events, especially in low-income or remote areas may go undocumented despite having significant local impacts."

Regarding the absence of European countries in Figure 5, we would like to clarify that this figure presents a sample representation of selected drought events globally and does not suggest that Europe did not experience droughts. In fact, CDI-detected events in Europe, such as the 2018 drought are discussed in the manuscript, and for further clarity, a sample map of European events is attached herewith (Figure 1). However, we also wish to highlight that, in comparison to other regions, Europe and Australia show fewer drought-related entries in the GDIS database. This supports our broader observation that developed regions (e.g., Europe, USA, Australia) tend to report fewer socio-economic drought events possibly due to stronger infrastructure, better preparedness, and adaptive capacity. This contrast further underscores the importance of evaluating agro-climatic indicators like CDI, which can identify stress conditions even when no disaster impacts are formally reported, especially in data-scarce or impact-resilient regions.

[Figure]

Figure 1. Comparison of CDI Detected Droughts and GDIS Events Over Europe: (a) 2017 Drought Over Italy with Strong CDI–GDIS Overlap, and (b) 2018 Drought Over Spain and Surrounding Regions Detected by CDI but Not Captured by GDIS.

4. In section 2.3 it is indicated that soil moisture is obtained using a weighting for each of the strata; however, it is not detailed how it is obtained, and the way in which this is done should be explained, since it is one of the variables used to characterise the combined drought index, and as shown in the weights in Figure 4, this variable is quite important in the determination of this combined factor.

Thank you for your comment. We appreciate the opportunity to clarify the method used to derive the soil moisture input for CDI, given its importance in the analysis. We used the ERA5-Land soil moisture dataset (European Centre for Medium-Range Weather Forecasts, 2023), obtained from the Copernicus Climate Data Store, covering the period 2001 to 2021 with a spatial resolution of $0.1° \times 0.1°$ and monthly temporal resolution. This dataset provides soil moisture values across four depth levels: (Layer 1: 0–7 cm, Layer 2: 7–28 cm, Layer 3: 28–100 cm, Layer 4: 100–289 cm). For our analysis, we used the first three layers (0–100 cm) to represent root-zone soil moisture, which is widely mentioned in the literature as critical for agricultural drought monitoring (e.g., Bolten et al., 2010; Entekhabi et al., 2010; Sehgal et al., 2017). Root-zone moisture reflects the water available for vegetation and crops and is particularly relevant for assessing impacts visible through NDVI and other surface stress indicators. To obtain a single representative soil moisture value for the top 1 m, we calculated a weighted average of the three layers based on their respective thicknesses. This approach ensures that deeper layers, which store more moisture and contribute more significantly to long-term water availability, are proportionally

represented, while still capturing the sensitivity of upper layers to short-term dry conditions.

We excluded the fourth layer (100–289 cm) from the analysis, as it extends well beyond typical rooting depths and is less responsive to seasonal surface drought, especially in relation to vegetation stress and agricultural impacts. The resulting root-zone soil moisture value was then standardized (e.g., using z-scores) and integrated into the CDI framework along with other drought-relevant indicators.

This explanation has been added in the revised manuscript under Section 2.3, lines 151-159, as follows:

"We used the ERA5 Land soil moisture dataset (European Centre for Medium-Range Weather Forecasts, 2023) acquired from the Copernicus Climate Data Store for the study period from 2001 to 2021. The monthly data products, with a spatial resolution of 0.1 x 0.1 degrees, were used for the study. The soil moisture datasets were available for different soil depth levels: first (0–7 cm), second (7–28 cm), third (28–100 cm), and fourth (100–289 cm). For our analysis, we used the first three layers (0–100 cm) to represent root-zone soil moisture, which is widely recognized in the literature (Bolten et al., 2010; Sawada, 2018; Sehgal et al., 2017) as critical for agricultural drought monitoring. To obtain a single representative value for soil moisture in the top 1 m, we employed a weighted averaging method using the respective thicknesses of the first three layers. The resulting weighted root-zone soil moisture layer was then standardized (e.g., using z-scores) and integrated into the CDI framework along with other drought-relevant indicators."

5. Section 2.5 states that '*we assumed January as the starting month and December as the end month of the respective event, and further analysis was carried out*'. Taking into account that the hydrological year in many databases is from October to September, It Would be important to know what percentage of the data used assume an unknown period, and to analyse the sensitivity of the results obtained to a possible alteration of the hydrological year.

Thank you for your insightful comment. In our dataset of 2,142 drought events derived from the GDIS database, 143 events (~6.7%) did not have a specified start month. For these events, we assumed January (month 1) as the default starting month to ensure their inclusion in the analysis.

We acknowledge that the hydrological year varies across regions. However, given that the number of events with missing start months represents a small proportion of the total dataset (<7%), we believe that this assumption has a minimal impact on the overall results. The number of such events and their percentage have now been explicitly mentioned in the revised manuscript under Section 2.5, lines 183-185 as:

"In some cases (143 events out of 2142 ~6.7%), due to the unavailability of monthly details in EM-DAT, we assumed January as the starting month and December as the end month of the respective event, and further analysis was carried out."

6. Regarding the results section, the results are presented in absolute terms by quantifying the number of detected and undetected events, which makes them difficult to understand. The quantification of the accuracy of the proposed methodology should be done using specific metrics obtained from a confusion matrix, detailing: Accuracy, Precision, recall, specificity, F1-score, AUC, etc. This will allow the discussion section to be completed by comparing similar metrics from previous work.

Thank you for your valuable feedback. In response to your comment regarding the

evaluation of our methodology using specific performance metrics, we have now included a detailed assessment of 'recall' (a key metric from the confusion matrix) across multiple scenarios. As shown in the new figure (below figure 2), we evaluated all five drought indices under four time windows: (a) Actual Event Period (AEP), (b) One Month Prior + AEP, (c) Two Months Prior + AEP, and (d) Three Months Prior + AEP.

We chose recall as a primary metric in this comparison because of its importance in drought detection, capturing as many actual drought events as possible is critical for early warning systems and risk mitigation. As such, higher recall values indicate better detection capability.

Across windows all the time, CDI consistently outperforms or is on par with other indices, especially when short droughts are excluded (i.e., longer droughts of ≥2 months). Example: In panel (a), CDI has the highest recall (0.94) when considering longer drought events. In panels (b), (c), and (d), CDI maintains recall values of 0.96–0.97, outperforming or matching the best-performing indices in each respective window. This shows the robustness and sensitivity of CDI in capturing drought events over both short- and long-term windows.

We agree that NDVI and SSMI show relatively high recall in specific cases (panel b, One Month Prior + AEP); such behavior is expected due to their sensitivity to vegetation stress and soil moisture. However, CDI demonstrates the most robust and consistent recall overall, across all scenarios and time windows, highlighting its superior ability to detect drought events under different conditions.

We also acknowledge the importance of evaluating performance through a complete confusion matrix, including false positives, false negatives, and derived metrics such as precision, specificity, and F1-score. However, this requires a complete and unbiased ground-truth dataset. As discussed in our response to a previous comment, GDIS does not capture all real-world drought events due to limitations in reporting infrastructure, particularly in data-scarce regions. As a result, metrics that rely on the assumption of full event coverage (such as precision or specificity) may be unreliable in this context. Therefore, we focused on recall as the most informative and robust metric for evaluating detection performance in a globally heterogeneous impact reporting system like GDIS.

However, your suggestion to explore additional performance metrics derived from the full confusion matrix is highly valuable and could be an excellent direction for future extensions of this study, particularly when more comprehensive ground-truth data becomes available.

This analysis and the newly generated figure have been added to the revised manuscript as follows (Section 4.3, lines 481–487):

For comparative analysis of drought detection performance, recall serves as a crucial metric, as it quantifies the ability of each index to correctly identify actual drought events. High recall is especially important in early warning systems where missing events can lead to unmitigated impacts. As illustrated in Appendix 7, the CDI index consistently outperforms others across all time windows, particularly for events lasting ≥2 months, where it achieves recall values between 0.94 and 0.97, demonstrating robust and reliable drought detection capability. For a more comprehensive understanding of detection performance, additional metrics derived from a full confusion matrix, such as precision, specificity, and F1-score, could provide further insights and represent a promising direction for future work.

[Figure]

Figure 2: Comparative recall performance of five drought indices (CDI, NDVI, SSMI, SPI, and STI) across different time windows and event durations.

7. From the visual analysis of Figure 7, it cannot be concluded that the combined index is clearly better than the individual indices, as indicated in line 525 as the results represented in that figure the combined index performs almost as well as the NDVI and in some cases slightly worse.

Thank you for pointing out this. We acknowledge that in some individual cases (as shown in Figure 7), other indices, such as NDVI or SSMI, show slightly higher event counts than CDI. This variation is expected because, NDVI and SSMI are highly sensitive to vegetation and soil moisture changes, which may capture localized drought signatures more prominently in certain periods.

However, our intention was not to claim that CDI outperforms all indices in every individual instance. Rather, the overall trend across different time windows and event durations demonstrates that CDI consistently maintains high performance, combining information from multiple indicators. This makes it more robust and generalizable across diverse drought conditions. We have revised the language in line 525 to better reflect this revised interpretation, in the revised manuscript (section 5: discussion, lines 606-609) as follows,

"The comparative analysis between the CDI and other individual parameter-based indices suggests that CDI offers a strong overall capability for detecting GDIS events, showing robust performance across time windows and a closer association with socio-economic impacts."

8. The analysis of the accuracy of the different indices could be completed with information on the socio-economic characteristics of each region as well as the typology of land cover in the region analysed, the accuracy metrics according to these variables in order to be able to conclude in a quantifiable way under which conditions one index performs better than another.

We thank the reviewer for the valuable suggestion. In response, now we have extended the analysis to include a zonal-level evaluation of index performance across the four major Köppen climate zones (Arid, Temperate, Tropical, and Cold). To carry out this analysis, initially GDIS polygons were spatially intersected with the respective climate zone shapefiles. However, due to topological limitations when working with two polygon layers, a direct one-to-one assignment was not always feasible. To address this, a 50% spatial overlap criterion was applied to assign each GDIS event to a climate zone. This ensured meaningful spatial classification and avoided ambiguous assignments. Applying this rule resulted in 2161 zonally attributed GDIS events, a slight increase from the original 2142 events due to partial overlaps at climate zone boundaries.

For each of these events, the start and end dates were used to extract corresponding values from multiple drought indices (CDI, SPI, SSMI, NDVI, STI). A threshold of -1 was used to determine whether an index detected a drought event. Based on this, index-wise detection counts were computed for each zone.

This analysis helps quantify the accuracy of each index under distinct climatological conditions, shedding light on their behavior under different topographies and regional drought patterns. The results are presented in the figure below.

We also thank the reviewer for suggesting the use of land cover typology and socio-economic characteristics. Incorporating these dimensions would be a valuable extension and offers a promising direction for future development of this research.

[Figure]

Figure 9. Zone-wise accuracy of drought indices (CDI, SPI, SSMI, NDVI, STI) in detecting GDIS events across four Köppen climate zones: Arid, Tropical, Temperate, and Cold. The bar heights represent the percentage of GDIS events accurately captured by each index (threshold = -1), and the numbers inside the bars indicate the absolute number of consistent detections. The total number of GDIS events considered per zone is: Arid – 571, Tropical – 949, Temperate – 453, Cold – 188 (Total = 2161).

This figure shows that while individual indices perform well under certain climate regimes (e.g., SSMI and NDVI in arid zones), the CDI consistently demonstrates high

association with GDIS events across all zones, suggesting its robustness and general applicability. This figure and analysis have also been included in the revised manuscript as Figure 9., and the detailed analysis is mentioned in section 4.3 lines 522 to 536:

"Figure 9 represents the zonal validation results for drought index performance across four climate zones (Arid, Tropical, Temperate, and Cold) based on Köppen's climate classification (threshold criterion: -1). The figure highlights how different indices perform differently under varying climatic conditions, while the CDI demonstrates consistently high detection accuracy across all zones. In the Arid zone (a), CDI detected 95.4% of GDIS events (the highest among all indices), while SSMI and NDVI also performed well with 93.0% and 90.5%, respectively. This can be attributed to the high sensitivity of NDVI and SSMI to vegetation and soil moisture stress, which are obvious under arid conditions. However, in the Cold zone (d), individual index performance dropped noticeably, with SPI detecting only 66.0% of events, whereas CDI still maintained 80.3% detection. Similarly, in the Temperate zone (c), both CDI and SSMI showed strong association with GDIS at 87.2% and 89.2%, respectively, indicating that some indices may be better suited for certain climate types. In contrast, Tropical zone (b) showed relatively lower detection percentages for all indices, with CDI still leading at 80.9%. SSMI performance in tropical and cold zones was lower, possibly due to dense vegetation cover and higher variability in surface moisture, which can limit the accuracy of soil moisture retrievals. These results emphasize that while individual indices can perform well in specific climate zones, their performance is not consistent across all zones. CDI, by integrating multiple indicators, offers more universally reliable detection, making it better suited for broader applications in drought monitoring across diverse climatic regions."

9. From line 470 to Figure 9, the purpose of testing the correlation between the combined index and the indices should be clarified. Considering that the combined index has been obtained from a principal component analysis of these variables, it is logical that it is correlated with the different parameters according to the weighting weights, it should be clarified what the purpose of this analysis is.

Thank you for this observation. We agree that, since the CDI is derived through PCA, a certain level of correlation with its input indices is expected due to the weighting structure. However, the purpose of this correlation analysis was threefold:

- To complement the spatially varying PCA weight maps by showing actual spatial alignment between CDI and its components over time. While the PCA weight maps reflect each component's contribution during index construction, they do not directly represent how consistently each index aligns with CDI behavior in practice.

- To account for the temporal variability of PCA weights. Since the PCA is performed monthly, the weights assigned to each component index vary over time. This makes it difficult to interpret long-term or spatial relationships between CDI and its components just by examining the weights. The correlation maps provide a more interpretable, time-integrated view of these relationships.

- To support the interpretation of CDI behavior across different climatic regions by identifying where specific indices (e.g., precipitation, NDVI, or SSMI) are more or less aligned with the composite CDI signal. This helps explain regional variations in CDI performance.

For example, Figure 9 shows that SSMI consistently has a strong positive correlation with CDI across most regions, emphasizing the dominant and stable role of soil moisture in

shaping the CDI signal globally.

To clearly convey the purpose of this analysis, we have included an explanatory statement in the revised manuscript (Section 4.3, lines 543–545):

"To better understand the spatial behavior of the CDI, a correlation analysis was performed (Figure 10) to examine how consistently each input index aligns with the composite signal across regions."

10. As mentioned above, there are conclusions that are not justified by the results presented. Lines 593 to 595, The following is stated: '*This novel index surpassed the performance of four commonly used single-parameter-based traditional indices, demonstrating superior accuracy in identifying GDIS droughts and effectively representing their socio- economic impacts*'. However, as mentioned above, this conclusion is not supported by the information and results presented in this work.

Thank you for pointing this out. We acknowledge that the original phrasing may have overstated the conclusion beyond what the presented results directly support. In the revised manuscript (lines 704–647), we have reworded the statement to more accurately reflect the findings. The revised text is as follows:

"The comparative analysis indicates that the proposed index performs consistently well across different drought scenarios and offers a more integrated representation of drought patterns, showing strong association with observed GDIS events and potential links to socio-economic impacts."

This revised conclusion is based on the recall-based performance metrics (Appendix 7), correlation patterns (Figure 10), and visual comparisons that demonstrate CDI's robustness across space and time. We believe this updated wording better aligns with the scope and evidence presented in the manuscript.

11. Lines 599 to 600: The following is stated: '*CDI-derived drought clusters exhibit a statistically significant representation of GDIS drought events (indicative of socio-economic impacts), with 95% of the GDIS events successfully 600 identified using the CDI*'. However, to support this assertion, it is necessary to consider the full set of metrics from the confusion matrix. While the index may identify GDIS-catalogued drought events, it could also detect other drought events that do not necessarily have socio-economic effects. Based on the results presented in this paper, it cannot be conclusively stated that the CDI index alone can determine the existence of socio-economic drought.

Thank you for this important clarification. We fully agree that detecting GDIS events alone does not confirm that the CDI can directly identify socio-economic droughts, as it may also detect events without documented impacts. Our intent was not to claim a deterministic relationship between CDI and socio-economic droughts, but rather to highlight the strong association between CDI-identified drought clusters and reported GDIS events.

To address this, we have revised the statement in the revised manuscript (section 6, line 712) to better reflect the scope of our findings. The updated sentence is as follows:

"CDI-derived drought clusters show strong spatial and temporal association with GDIS-reported drought events, with approximately 95% of GDIS events successfully identified using the CDI. This suggests that the index effectively captures drought conditions that frequently align with documented socio-economic impacts."

A full confusion matrix requires reliable identification of false negatives, that is, actual

socio-economic drought events that were not captured by GDIS. As discussed in earlier responses, this is not currently feasible due to the incomplete coverage of real-world drought impacts in GDIS. Therefore, our analysis is limited to comparing CDI signals against known, documented events and does not aim to establish CDI as a comprehensive socio-economic drought detector. Since there is no globally consistent and exhaustive database of socio-economic drought impacts, we cannot objectively determine which CDI-identified events are true false negatives or simply unreported impacts, making such analysis infeasible at this stage. However, developing such a reference dataset and enabling robust false negative analysis would be a valuable direction for future research.

**Technical corrections**

1. It should be clarified whether the ranges in Figure 3 correspond to the classification set out in Figure 1, and if so, the nomenclature should be homogenised.

Thank you for this observation. We clarify that the ranges in Figure 3 do not correspond to the classification presented in Figure 1. Figure 1 displays the CDI classification scheme, where values range from negative to positive, representing a range from wet to drier drought conditions. On the other hand, Figure 3 presents the frequency of drought events, using a color gradient from light yellow to dark red to indicate low to high drought frequency. While some of the color tones may appear visually similar between the two figures, they represent entirely different variables and scales. We have revised the figure caption for Figure 3 to clarify this distinction. The revised caption is as follows,

"Figure 3. Spatial distribution of GDIS drought frequencies: (a) Global scale, (b) East Africa, (c) America, and (d) Asia. The drought frequencies range from one to eight, represented by shades from light yellow (low frequency) to dark brown (high frequency). Note: The color scheme used here is distinct from the CDI classification shown in Figure 1 and represents event frequency, not drought intensity."

2. Figure 7 has very low resolution, however, as it is proposed to replace the figure with a display of the results in terms of the metrics of the confusion matrix, I understand that this figure will be replaced in its new format.

Thank you for pointing this out. In the revised manuscript, Figure 7 has been replaced with a higher-resolution image to improve visual clarity. Additionally, as suggested, a confusion matrix–based recall analysis has been added as a new figure in Appendix 7 to provide a more quantitative comparison of index performance.

********************* Thank You *********************

---

## Author Comment (AC3)

**Response to Reviewer 3 Comments**

**MS No.: hess-2024-245**

Thanks to the reviewer for reviewing our manuscript. We sincerely appreciate the time and effort you dedicated to carefully evaluating our work and providing valuable comments and suggestions. Your insights have significantly contributed to improving the quality of our manuscript. We have addressed each of your comments accordingly, and our responses are provided below. Reviewer comments are highlighted in red, while our responses are in black.

This study presents a global assessment of combined drought indicator (CDI) in identifying socio-economic drought impacts at the subnational level and compares it with single hydro-meteorological drought indices, such as SPI, STI, NDVI, and SSI. The GDIS dataset was utilized to represent observed socio-economic drought impacts. The authors found that CDI outperforms any single drought index in detecting drought events. Its performance is even higher when short drought events lasting 2 months or less are excluded in the analysis. This study highlights the importance of using CDI to evaluate socio-economic drought impacts.

**Assessment**

This paper analyzes the performance of CDI in identifying drought events recorded in the GDIS database. Furthermore, the study compares the performance of CDI with individual hydro-meteorological drought indices in detecting droughts. I find the work and its findings interesting, especially when I look at the title, which suggests an assessment of socio-economic drought impacts. However, the aim and title of the manuscript seems misleading. Additionally, the manuscript contains some typos and formatting issues. Below, I provide four general comments, aimed at improvement and clarification. I suggest the authors consider these comments in their revised manuscript.

**General Comments**

I have four general comments regarding the manuscript:

1. The aim of this paper is to evaluate the performance of multiple drought indices, including CDI to pinpoint drought areas recorded in the GDIS database. This objective is also stated in the abstract. However, the title suggests an assessment of socio-economic drought impacts using both single and combined drought indicators. When I read the paper, I do not find any results related to drought impacts; instead, the analysis focusses solely on drought occurrences. I expected to see socio-economic impacts, such as economic loses, human mortality, etc. Therefore, I suggest either incorporating an analysis of socio-economic impacts or modifying the title to better reflect the study's aim and findings. I personally make a clear definition between event and impact.

We thank the reviewer for the helpful comment and for suggesting a title change based on the analysis and results presented. We agree that the original title may have conveyed a misleading impression that our study analyzes socio-economic drought impacts, such as economic losses or

human health consequences, whereas our focus is specifically on drought events as recorded in the GDIS database. Following the reviewer's suggestion, we have revised the title to better reflect the actual scope of our work. The updated title is: **"Global Assessment of Socio-Economic Drought Events at Sub-National Scale: A Comparative Analysis of Combined Versus Single Drought Indicators."**

By using the term "events" rather than "impacts," we aim to clearly distinguish that our analysis concerns to the detection and characterization of recorded drought occurrences, not their quantified impacts. This distinction aligns with the reviewer's own helpful clarification between event and impact, which we found valuable in refining the manuscript.

The introduction section could be better structured for improved readability and logical flow. Currently, the introduction negins with a general statement about droughts (paragraph 1), followed by socio-economic drought impacts (paragraph 2), GDIS database (paragraph 3), and then go back again to drought types and indicators (paragraph 4), and finally discusses the CDI (paragraph 5). To enhance the coherence, I suggest reorganizing the introduction so that the discussion on droughts (paragraphs 1, 4, 5) comes first and then followed by impacts and the impact database (paragraph 2 and 3). By doing this, it would create a more logical and improve the overall flow of the paper.

We thank the reviewer for the suggestion regarding the structure of the introduction section. As recommended, we have reorganized the content to improve logical flow and coherence. The revised introduction now begins with a discussion on drought types, indicators, and the CDI, followed by sections on socio-economic drought impacts and the GDIS database. The updated introduction section is as follows (revised manuscript, section 1, line 35-130):

[revised manuscript text omitted]

2. If the GDIS database includes different types of drought impacts, such as economic loses, human mortality, forest fire, yield loses, etc, I suggest establishing a link between drought indicators and their corresponding impacts. For example, if NDVI is the selected indicator for a certain region, I expect that agricultural impact is the dominant impact in that region. If the selected indicator is SPI, then maybe the dominant impact on that region might be related to meteorological drought, and so on. Or at least, I recommend discussing these relationship in the paper.

Thank you for this thoughtful suggestion. While the GDIS dataset includes general information such as the number of affected people and fatalities, it does not consistently categorize the type of socio-economic impact (e.g., agricultural loss, mortality, economic damage) in a structured format. Therefore, this study does not attempt to link specific impact types with particular drought indicators. Instead, the focus is on evaluating whether drought events identified by CDI and other indices align with GDIS-reported events that reflect broader socio-economic consequences. We agree that future research could benefit from exploring such impact-specific relationships where detailed data are available.

3. Regarding drought indicators, the accumulation periods used for analyzing SPI, STI, and SSI are not clearly stated. I suspect that the authors have only used 1 month accumulation period. I suggest reconsidering the use of the term SSI for soil moisture drought. In many publications, SSI refers to the Standardized Streamflow Index, which is used to identify streamflow drought. The standardized soil moisture index is commonly referred as SMI or SSMI.

We thank the reviewer for pointing this out. Yes, we confirm that a 1-month accumulation period was used for analyzing SPI, STI, and SSI in this study. The choice of a 1-month timescale was intentional, as it has several advantages for our analysis. Specifically, it is more sensitive to short-term changes, capturing rapid variations in agro-climatic variables. This is important for assessing early-stage drought conditions and understanding drought impacts triggered by short-term events. Additionally, it aligns with the GDIS timescale, as GDIS provides data on a monthly level, which helps in more accurate event-level analysis. Several previous studies were also referred, that support the use of a 1-month timescale for capturing short-term drought dynamics.These details have also been included in the revised manuscript (section 3.1, line 213 to 217) as follows:

"In this study, a 1-month accumulation period was used for SPI, STI, and SSMI to capture short-term drought dynamics. This timescale is sensitive to rapid changes in agro-climatic conditions and supports early-stage drought detection. It also aligns with the monthly resolution of GDIS data, enabling accurate event-level analysis. The choice enhances consistency between drought indicators and the event database."

Thank you for the suggestion and for highlighting the potential confusion with the abbreviation "SSI." In the revised manuscript, we have updated the terminology to "SSMI" to refer to the Standardized Soil Moisture Index, which is more commonly used in the literature.

**Line by line comments**

L refers to line and P refers to page.

**P1**: Abstract: I suggest to expand the abstract in order to capture the summary of your study. I missed an explanation about PCA and results on long and short drought durations. In the EGU journal, you can have longer abstract.

Thank you for this suggestion. The abstract has been expanded in the revised manuscript as follows:

"The accurate assessment of the propagation of drought hazards to socio-economic impacts poses a significant challenge and is still less explored. To address this, we analyzed a sub-national disaster dataset called the Geocoded Disaster (GDIS) and evaluated the skills of multiple drought indices to pinpoint global drought areas identified by GDIS. For the comparative analysis, a widely used Standardized Precipitation Index (SPI), Normalized Difference Vegetation Index (NDVI), Standardized Soil Moisture Index (SSMI), and Standardized Temperature Index (STI) were globally computed at the subnational level for the period 2001–2021. Out of 1641 drought events recorded in GDIS, NDVI identified 1541 (93.9%), SPI 1458 (88.8%), STI 1439 (87.7%), and SSMI 1376 (83.9%) events, respectively. NDVI showed better performance in highly vegetated

areas due to its sensitivity to precipitation and soil moisture and its inverse relationship with temperature.

Recognizing the limitations of single-input drought indices in capturing the complex propagation of droughts, we also introduced a novel Combined Drought Indicator (CDI), which integrates meteorological (rainfall and temperature), and agricultural (NDVI and Soil moisture) anomalies using a weighted approach to identify droughts and play a key role in minimizing inaccuracies in drought assessment. CDI successfully identified 1550 (94.5%) of the GDIS documented drought events, outperforming all individual indices. Based on CDI, the highest frequency of severe droughts (>7 events) was observed in sub-Saharan Africa and South Asia. It also captured persistent droughts in Argentina, Brazil, the Horn of Africa, western India, and north China; areas that are highly vulnerable to socio-economic droughts. Our findings highlight the importance of using CDI for improved identification of socio-economic drought events and for prioritizing regions at greater risk."

**P1L12**: In this sentence, the study area should be mentioned, which is global. I only found this in Line 15.

Thank you. As suggested, the study area (global) has now been added to the third line of the revised manuscript: '…evaluated the skills of multiple drought indices to pinpoint global drought areas identified by GDIS.'

**P2L36**: Here the authors can see an example of reference typo.

Thank you. The referenced typo has now been corrected to (Hossain et al., 2022).

**P2L40**: In the last sentence of paragraph 1, the authors mention about drought propagation from meteorological drought to socio-economic drought. However, I miss an explanation on drought types. Later in paragraph 4, I read meteorological drought, agricultural drought, and hydrological drought.

In the revised manuscript, the introduction section has been reorganized. Hence, the missing explanation gap has now been answered.

**P2L48**: Again, I see typo on references -> Tiwari and Mishra (2019) "and" Wu et al. (2018).

Thank you. The referenced typo has now been corrected to : Tiwari and Mishra (2019) and Wu et al. (2018)

**P2L52**: Please provide references for DIR and EDII.

Thank you for this suggestion. The references have been included in the revised manuscript as follows;

DIR : (National Drought Mitigation Centre, 2025) : Drought Impact Reporter (DIR) | Drought.gov: https://www.drought.gov/data-maps-tools/drought-impact-reporter-dir, last access: 26 March 2025.

EDII : European Drought Centre, 2025): European Drought Impact Report Inventory (EDII) and European Drought Reference (EDR) database - European Drought Centre: https://europeandroughtcentre.com/news/european-drought-impact-report-inventory-edii-and-european-drought-reference-edr-database/, last access: 26 March 2025.

**P2L54**: Again, missing comma between references.

Thank you for pointing out this. In the revised manuscript, all the references are correctly cited.

**P2L59-63**: I suggest the authors to explain more about GDIS. Or at least describe GDIS in detail in the method section.

Thank you for the suggestion. We agree that a more detailed description of GDIS improves clarity. We have expanded the Data section 2.5 to provide additional information on the spatial structure, content, and relevance of GDIS, including its advantage over EM-DAT in capturing subnational impact data. The revised explanation in the manuscript (Section 2.5, lines 167–176) is as follows:

"The Global Disaster Identifier System (GDIS) dataset is a geocoded dataset distributed by SEDAC (NASA) that links disaster events to specific administrative boundaries (Rosvold and H. Buhaug., 2021). It builds upon the EM-DAT database by adding spatial GIS information, providing polygons for subnational regions affected by specific disaster events. Each event is tagged with the type of hazard (e.g., drought, flood, cyclone), the time period, and the affected regions at the sub-national level. In this study, only drought events were extracted from GDIS for the years 2001 to 2021. The GDIS dataset is based on the EM-DAT dataset. EM-DAT records a natural disaster when it meets any of the following conditions: 10 or more fatalities, impacts 100 or more individuals, or prompts a declaration of a state of emergency along with a request for international aid.

The use of GDIS, instead of relying solely on EM-DAT, adds spatial specificity and improves the ability to evaluate drought impacts in different regions, especially in developing countries where vulnerability is high. However, it's worth noting that GDIS coverage is not globally uniform and may be affected by underreporting in certain regions."

**P3L77**: Missing word "and" between Sandeep et al. (2021) and Tao et al. (2021).

Thanks. The references have been corrected and updated in the revised manuscript.

**P3L91**: Missing space between parameter and (Jiao et al., 2019a). Starting from here, I will not mention one by one the typo regarding missing space or in references. Please check carefully throughout the manuscript.

Thank you for pointing out the formatting errors. In the revised manuscript, a space has been added between 'parameter' and '(Jio et al., 2019a)'. The rest of the write-up has also been carefully checked for formatting and typographical errors.

**P3L95-97**: European Drought Observatory (EDO) also utilizes combined drought indices.

Thank you for this information. This reference have been included in the revised manuscript (line 95).

EDO: (European Environment Agency, 2025) : European Drought Observatory (EDO) — European Environment Agency: https://www.eea.europa.eu/policy-documents/european-drought-observatory-edo, last access: 26 March 2025.

**P4L122**: Here, it is clearly stated GDIS drought events and not impacts.

Thank you for your comment. Yes, it is a GDIS drought event, not an impact. In the revised manuscript, we have carefully distinguished between the terms 'impacts' and 'events' and used them appropriately.

**P5L139**: Better use x for indicating resolution.

Thanks. We agree with your comment, and in the revised manuscript, to indicate resolution x has been used instead of *.

**P5L141**: In this sentence, the authors mention: to explore the linkage between drought indicators and "socio-economic impacts". It is drought impacts or drought events? It is very confusing. I make a clear distinction between event and impact (see my general comment).

Thank you for this comment. We agree that it is important to distinguish clearly between drought events and their impacts. In the context of this study, we refer specifically to GDIS events, which represent drought events that have been recorded due to their associated socio-economic impacts. We have revised the manuscript wording to clarify this distinction and avoid confusion between the terms "event" and "impact."

**P5L152**: The authors may remove the word "originally" since ERA5 Land already has spatial resolution of 0.1 degree.

Thank you. In the revised manuscript, we have removed the word 'originally' when referring to the spatial resolution of ERA5.

**P6L170**: Define what is socio-economic impact of drought in the GDIS data.

Thank you for this comment. In the GDIS dataset, a socio-economic impact of drought is considered to have occurred when a drought event meets one or more of the EM-DAT disaster inclusion criteria. (i) it causes at least 10 deaths, (ii) affects 100 or more people, (iii) leads to a state of emergency declaration, or results in a call for international assistance. These criteria reflect

real-life impacts such as food insecurity, loss of livelihoods, reduced agricultural output, water shortages, or displacement. We have added this clarification to the revised manuscript (Section 2.5).

**P6L175**: Disaster no or disaster number?

Thank you for pointing this out. We have corrected it to 'disaster number' instead of 'disaster no.' in the revised manuscript.

**P8L232-233**: I think *i*th and *j*th should be in italic.

Thanks for the comment. In the revised manuscript, ith and jth have been written as *i*-th and *j*-th.

**P9L253-254**: How about if drought indices indicate drought but GDIS not, so is it false alarm?

Thank you for this question. Yes, we did find a few instances where drought indices indicated drought severity, but these were not detected by GDIS. For example, the CDI detected severe drought in South Argentina during 2014–15 and in Namibia in 2013. Additionally, the CDI identified drought conditions over parts of Europe in 2018 that were not reflected in the GDIS data. These discrepancies may be due to adaptation techniques or practices implemented in these regions, which effectively managed agro-climatic extremes or hazards without significantly affecting local socio-economic conditions. Additionally, it is important to note that GDIS does not fully capture all drought events globally, particularly in regions with limited disaster reporting infrastructure or under-documented local impacts. This explanation has been included in the revised manuscript in Section 4.2, lines 434-439 as follows:

"CDI detected severe drought events in South Argentina during 2014–15, Namibia in 2013, and parts of Europe in 2018, which were not reflected in GDIS event records. These instances highlight that; not all agro-climatic droughts lead to recorded socio-economic impacts, especially in regions with strong adaptation and mitigation capacities. Practices such as advanced irrigation, drought-resistant crop varieties, or effective early warning systems may help manage the agricultural and societal impacts of climatic stress, thereby reducing the likelihood of such events being recorded in GDIS."

**P13**: Figure 4, please provide alphabets a, b, c, and d for each figure in Figure 4.

Thank you for the suggestion to add sub-details to the figure. This will indeed help readers better understand the content. In the revised manuscript, we have included captions for each subfigure. Additionally, we have modified the figure title to include sub-numbering. The updated figure and its title are as follows;

[Figure]

Figure 4. Example of pixel-based weights for the four input variables of CDI: rainfall (a), LST (b), soil moisture (c), and NDVI (d), calculated using the PCA method for a sample month (April). The weights range from 0 to 1, with colors varying from dark red (lower weights) to dark green (higher weights).

**P14L350**: Same as Figure 4, please provide alphabets for Figure 5. Here the authors refer to Figure 5a but there is no figure 5a.

As suggested by the reviewer, sub-details (alphabet labels) have been added to Figure 5, similar to Figure 4, in the revised manuscript. The updated figure and its title are as follows:0

[Figure]

Figure 5. Assessment of Drought Using CDI with Overlay of GDIS Events over America (June 2015) (a), Africa (June 2009) (b), and South America (August 2009) (c). The black polygons

represent GIS polygons from GDIS, indicating drought-affected administrative units based on GDIS data. The base maps display CDI results for the respective GDIS drought months, ranging from dark brown (indicating extreme dry conditions) to dark green (indicating extreme wet conditions). The alignment of GDIS polygons with droughts detected by CDI demonstrates the CDI's capability to accurately identify GDIS droughts during the respective periods.

**P14L355**: What is AEP?

Thank you. AEP has been used as an abbreviation for Actual Event Period. Since the full form was already introduced in Table 2, we referred to it as AEP in this section. However, in the revised manuscript, we have also added the full form of AEP in the main text for clarity.

**P14L360**: The authors can also explain about false alarm. Furthermore, Table 2 is summarizing the findings and it is worth detailed explanation, such as the results with and without short droughts and the shifting.

Thank you for raising the point about false alarms. We did observe instances where CDI detected drought conditions that were not reflected in GDIS, and vice versa.

Despite the higher association of CDI in detecting GDIS events, there were a few instances (e.g., Burundi, and parts of Thailand) where CDI failed to capture GDIS events. The smaller spatial extents (subnational scales in Southeast Asia or parts of Africa) could be one reason for this discrepancy. Additionally, these regions might be experiencing other types of stress beyond agro-climatological factors (as indicated by CDI) that contribute to the GDIS events. It was also found that CDI identified stress conditions in some locations that were not reflected in GDIS. For instance, CDI detected severe drought in South Argentina during 2014-15 and Namibia in 2013. Additionally, CDI identified drought conditions over parts of Europe in 2018 that weren't reflected in GDIS data. These disparities may arise from adaptation techniques or practices implemented in these regions, effectively managing agro-climatic extremes or hazards without significantly impacting local socio-economic conditions. The same detailed reasoning behind these discrepancies is provided in the revised manuscript, Section 4.2, line 385

We agree that not all the details from Table 2 were explained in the text earlier. However, in the revised manuscript (Section 4.2), we have now included those results. The detailed findings are as follows;

In the first criterion, when considering one month prior to the AEP and using a threshold of -1, CDI detected 1,954 events, accounting for 91.22% of the total. When the threshold was changed to 0, 2,130 events (99.44%) were identified by CDI in alignment with GDIS. When short-duration drought events were excluded under the same criterion, CDI detected 1,573 events (95.86%) at the -1 threshold and 1,637 events (99.76%) at the 0 threshold. Under the second criterion, two months prior to AEP, CDI detected 2,010 events (93.84%) using a threshold of -1 and 2,137 events (99.77%) when the threshold was adjusted to 0. Excluding short drought events in this scenario, CDI identified 1,587 events (96.71%) at the -1 threshold and 1,637 events (99.76%) at the 0 threshold. In the final criterion, with considering three months prior to AEP, CDI detected 2,042 events (95.33%) at the -1 threshold and 100% of events at the zero threshold. After excluding short

drought cases, 1,589 events (96.83%) were captured at the -1 threshold, and again, all events (100%) were identified at the 0 threshold. These results highlight that GDIS events, which appear to occur in specific months, may actually be the outcome of agro-climatic variability occurring one, two, or even three months prior to the reported event, rather than being confined to that specific month alone.

**P15L376**: Beside the number, I think it is more meaningful to also present the percentage. How many percent drought was detected.

Thank you for this suggestion. We agree that adding percentages makes the analysis more meaningful and easier to understand. In the revised manuscript, percentages have been incorporated into the written analysis, and all relevant percentages have also been added to Table 2.

**P16L409**: Lower threshold here means index below 0 or long drought duration? I think the latter.

Thank you for the question. In our analysis, the term 'lower threshold' refers to the CDI index value used to detect drought events (-1 or 0). A threshold of -1 is stricter, meaning it represents more severe drought conditions, while a threshold of 0 is more relaxed. This does not refer to drought duration.

**P17L426-433**: In this paragraph, the authors could highlight which shifting month yields highest performance and false alarm.

Thank you for the suggestion. Based on our analysis, the three-month prior shifting criterion, particularly with a threshold of 0, yielded the highest detection performance, capturing 100% of the GDIS events. However, this setting may also introduce a higher possibility of false alarms, as events detected further in advance may not always correspond to actual socio-economic impacts. Conversely, the one-month prior shift with a threshold of -1 showed lower detection rates (88%), suggesting fewer false alarms but also reduced sensitivity. We have now highlighted these observations in the revised manuscript (section 4.2) to better clarify the relationship between early detection and the risk of false positives. These observations are mentioned in section 4.2, line 395 as follows:

"Among the shifting windows, the three-month prior criterion with a threshold of 0 demonstrated the highest detection rate, capturing 100% of GDIS events. However, this also suggests a higher likelihood of false alarms, as early indicators may not always align with actual drought impacts."

**P19**: Figure 8. The authors could make a bold line for threshold -1 in order to improve the readability.

Thank you. As per the reviewer's suggestion, the threshold line at -1 has been bolded in Figure 8 to enhance readability.

**P19L471-472**: Where can I see the figures showing the correlation between CDI and hydro-meteorological indices? The authors could provide these figures in the Appendix.

Thank you for the suggestion. In the revised manuscript the correlation results for a sample month (April) are presented in Figure 9. This figure illustrates the spatial correlation between CDI and key hydro-meteorological indices, including SPI, STI, NDVI, and SSMI.

[Figure]

Figure 9. Spatial Correlation Between CDI and Single Input-Based Traditional Indices for a Sample Month (April): (a) CDI vs. SPI, (b) CDI vs. STI, (c) CDI vs. NDVI, and (d) CDI vs. SSMI. Negative correlations are represented by shades from yellow to red, while positive correlations are shown in shades from light green to dark green.

**P21L501-502**: Make it clear that SPI-1 and SPI-3 are effective to detect meteorological and agricultural droughts while longer time scales are better for detecting hydrological droughts.

Thank you for your valuable suggestion. We agree with this observation and have revised the manuscript accordingly. The updated text now clarifies that SPI-1 and SPI-3 are typically used to detect meteorological and agricultural droughts, respectively, while longer time scales such as SPI-6 or SPI-12 are more appropriate for identifying hydrological droughts. This clarification has been included in Section Five (discussion), Lines 574-580 of the revised manuscript. The revised text is as follows;

"Previous studies, whether regional or global, often relied on single-parameter-based indices for drought monitoring, such as SPI (Ji and Peters, 2003; Liu et al., 2022; Mckee et al., 1993). Each of these has limitations for understanding drought. For instance, SPI is commonly used, but its effectiveness depends heavily on the selected time scale. Shorter time scales, such as SPI-1 and SPI-3, are effective for detecting meteorological and agricultural droughts, respectively, while longer time scales (e.g., SPI-6 or SPI-12) are more suitable for identifying hydrological droughts. However, in regions with distinct wet and dry seasons, SPI can sometimes misrepresent actual

drought conditions, for example, a short-term SPI may indicate drought during a naturally dry season, even when annual water availability remains within normal limits."

**P21L530**: Or maybe the impact is not related to meteorological drought impact?

Thank you for the insightful comment. We agree that the observed impact on the Horn of Africa may not be solely attributed to meteorological drought. The GDIS-detected drought event reflects agricultural or hydrological drought conditions, which are not fully captured by SPI. In response to your suggestion, we have revised the paragraph to include this explanation.

The updated text in the revised manuscript (Section 5: Discussion, Lines: 610-616) is as follows:

*"One such inconsistency was observed in the Horn of Africa, where the CDI and other indices identified a GDIS drought event, but the SPI did not. This discrepancy could be due to less pronounced precipitation anomalies in the SPI for this region, where baseline rainfall is already low and may not reflect drought conditions accurately. Additionally, it is possible that the observed impact was not solely related to meteorological drought but rather to agricultural or hydrological drought. Factors such as reduced soil moisture, land degradation, or soil types with low water retention capacity, captured by SSMI, STI, NDVI, and CDI, may have played a more significant role in triggering the event."*

**P21L533**: Same, maybe the impact is not related to soil moisture drought impact.

Thank you for the helpful comment. We agree that the drought impact in North Argentina may not have been primarily related to soil moisture conditions. While the presence of the Paraná River could buffer soil moisture variability, the observed drought event might have been driven by meteorological factors such as reduced rainfall or elevated temperatures, which may not be captured by SSMI. In response, we have revised the manuscript to include this alternative explanation.

The updated paragraph in the revised manuscript (Section 5: Discussion, Lines: 616-621) is as follows:

"Another example is from North Argentina (South America), where the SSMI failed to detect a drought that was identified by other indices. This disparity might be due to the presence of the Paraná River, the second-largest river in South America, which provides a significant source of soil moisture. Therefore, the SSMI might not reflect drought conditions. However, it is also possible that the impact was not directly related to soil moisture but instead resulted from meteorological factors such as reduced rainfall or elevated temperatures, which could lead to drought and socio-economic stress, as reflected in GDIS."

**P23L571:** Do the authors mean takeaway message?

Thank you for the comment. Yes, we wanted to convey that the association between CDI and GDIS is one of the key takeaway messages of the study. To improve clarity, we have revised the sentence in the manuscript.

The updated sentence now reads: "The association between the CDI and the GDIS is one of the key takeaway messages of our study, as it directly addresses the gap in understanding actual drought hazards and their socio-economic impacts."

**P23L580-581**: I think the statement here is very weak saying that lack of availability of high-res hydrological data. There are some high-res hydrological models that provide hydrological data. The Lisflood hydrological model of Europe provide 1x1 km spatial data for streamflow and soil moisture. Or do the authors mean in situ observation?

Thank you for this helpful comment. We agree that there are high-resolution hydrological datasets available from models such as LISFLOOD, particularly for specific regions like Europe. Our original intent was to highlight the limited global availability of consistent, high-resolution hydrological data, whether observed or modeled that can be used uniformly across diverse regions. While regional models exist, their spatial coverage and consistency vary globally, which poses challenges for global-scale analysis. We have revised the manuscript (lines 683-688) to clarify this point.

The updated sentence: "Due to the limited availability of consistent and high-resolution hydrological data at the global scale, hydrological variables were not included in this study. This may have contributed to certain disparities in detecting GDIS events. While regional high-resolution modeled datasets (e.g., LISFLOOD in Europe) are available, the lack of globally consistent and validated hydrological data remains a constraint. In future work, we will aim to incorporate hydrological variables where feasible and explore alternative methods for computing CDI weights to further improve drought detection and its linkage to socio-economic events."

**P23L592-593**: Mention again the agricultural and meteorological drought indicators.

Thank you for the suggestion. In the revised manuscript, meteorological and agricultural variables have been explicitly mentioned again in the conclusion section line 703. The updated sentence: "To address this limitation, we developed a new Combined Drought Indicator (CDI) that integrates two agricultural (Soil Moisture and NDVI) and two meteorological (Rainfall and Temperature) variables to assess drought conditions."

**P24L601-602**: Does GDIS provide drought socio-economic impacts or just drought events? (see my general comment). Please make it clear.

Thank you for the comment. We acknowledge the need for clarification. GDIS provides information on drought events, not direct socio-economic impacts. This distinction has now been clearly stated in the revised manuscript (Section 6: conclusion, Line 715). The updated sentence:

"The GDIS dataset provides direct socio-economic hazard event information and can be a valuable validation tool for drought indices."

Thank you for pointing out this oversight. In the revised manuscript, Figure g has been added to Appendix 1 to correct the error. The updated figure is as follows:

[Figure]

Appendix 1. Significant drought events detected by CDI in various global regions include: (a) the USA in August 2007 and (b) July 2012, (c) the Horn of Africa in July 2015, (d) Malawi and Zambia in February 2016, (e) western and northern China in July 2015, (f) India in July 2015, and (g) Australia in January 2019.

**P33**: Provide alphabetical figures.

Thank you for your comment. In the revised manuscript, the figures in Appendix 3 have been arranged alphabetically. The updated figure is as follows.

[Figure]

Appendix 3: Spatial locations-wise performance of CDI (a), SPI (b), NDVI (c), SSMI (d), and STI (e) in detecting GDIS events, where consistent events detected by each index are shown in pink, while inconsistent GDIS events (not observed) are shown in dark brown. The drought identification criteria were set to events during the actual drought period without considering short droughts (total GDIS events count = 1641).

********************* Thank You *********************

---

## Author Response (AR2)

Response to Referee 3 Comments

**MS No.: hess-2024-245**

We would like to express our sincere thanks to Dr. Sutanto Samuel Jonson for reviewing our manuscript once again. We deeply appreciate the time and effort you dedicated to thoroughly evaluating the revised version and offering thoughtful and constructive feedback. Your continued insights have played a crucial role in further enhancing the quality of our work. We have carefully addressed all your comments in this revised manuscript, with our detailed responses provided below. Reviewer comments are highlighted in red, and our responses are shown in black.

**General Comments**

I appreciate the authors' careful consideration of my feedback in their revised manuscript and improved the clarity of the paper. I only have minor textual comments that could be considered when submitting the final version. I accept the manuscript after corrections.

L refers to line

1. L49: Move the reference to the end of the sentence.

Thank you for the suggestion. As advised, the sentence has been revised to move all references to the end. The updated sentence is as follows:

"For example, the SPI has been used to assess droughts in Greece, the United Kingdom, Iran, India, and China by  Bhunia et al. (2020), Blain et al. (2022), Kazemzadeh et al. (2022), Livada and Assimakopoulos (2007), Zhang et al. (2012), respectively."

2. L94: EDO is agricultural drought monitoring and therefore it is not impact monitoring.

Thank you for pointing this out. In the revised manuscript, EDO has now been correctly referred to as a system for agricultural drought monitoring. The revised sentence is as follows;

"The European Drought Impact Report Inventory (European Drought Centre, 2025), which monitors drought impacts, and the European Drought Observatory (European Environment Agency, 2025), which monitors agricultural drought conditions, provide region-specific insights, but lack global coverage."

3.L96-98: Better combine these sentences with the next paragraph about GDIS.

Thank you for the suggestion. We have now merged the previously separated sentences into a single, coherent paragraph about GDIS as advised.  The updated text is as follows:

"Recently, the Geocoded Disaster (GDIS) dataset has been developed based on EM-DAT, offering geocoded disaster locations at a subnational level (Rosvold and H. Buhaug, 2021), along with detailed data on affected populations, fatalities, and economic losses. By addressing the limitations of EM-DAT, the GDIS dataset provides detailed information on socio-economically affected areas and administrative units in GIS polygon format. This spatially explicit dataset enables analysis of drought impacts across diverse socio-economic contexts. In this paper, we used this newly developed GDIS dataset and show that it enables us to explore the less-understood link between drought hazards and their socio-economic repercussions more accurately and comprehensively."

4.L161: Please provide reference for NDVI from Modis.

Thank you. As suggested, the reference for NDVI has been added in the revised manuscript as follows:

"We used monthly NDVI data products from the Moderate Resolution Imaging Spectroradiometer (MODIS) (Didan and Huete, 2023) spanning from 2001 to 2021."

5. L171-172: Redundant statement about GDIS. Maybe remove?

Thank you for pointing this out. We agree that the sentence was redundant and have removed it from the revised manuscript. The deleted sentence was:

"The GDIS dataset is based on the EM-DAT dataset."

6. L195: Table 1 could be improved by providing columns for e.g., data, description, spatial resolution, and sources.

Thank you for the suggestion. Table 1 has been revised to improve clarity and organization by adding separate columns for data type, description, spatial resolution, and source. The revised Table 1 is as follows:

**Table 1.** Details of the datasets used for this study.

| Data | Description | Spatial Resolution | Source |
|---|---|---|---|
| Rainfall | CHIRPS rainfall data | Original: 0.05° × 0.05°
 Resampled: 0.1° × 0.1° | https://www.chc.ucsb.edu/data/chirps |
| Temperature | ERA5-LAND monthly temperature | 0.1° × 0.1° | https://cds.climate.copernicus.eu/cdsapp#!/dataset/reanalysis-era5-land-monthly-means |
| Soil Moisture | ERA5-LAND monthly volumetric soil moisture | 0.1° × 0.1° | https://cds.climate.copernicus.eu/cdsapp#!/dataset/reanalysis-era5-land-monthly-means |
| NDVI | MODIS NDVI (MOD 13A3 product) | Original: 1 km × 1 km
 Resampled: 0.1° × 0.1° | https://modis.gsfc.nasa.gov/data/dataprod/mod13.php |
| GDIS | Geocoded Disaster dataset based on EM-DAT, event-wise socio-economic impact data | Spatial: Subnational
 Temporal: Event-wise | https://sedac.ciesin.columbia.edu/data/set/pend-gdis-1960-2018/data |

**7. L230: Please provides references for PCA when you said widely used.**

Thank you for the suggestion. We have added supporting references to justify the statement regarding the widespread application of PCA in atmospheric and hydrological studies. The revised sentence in the manuscript is as follows:

"The Principal Component Analysis (PCA) technique was used to assign weights to all four input indices. PCA has been widely used in atmospheric and hydrological studies to describe dominant patterns in multivariate data (Anon, 2002; Hannachi et al., 2007; Jackson, 1993)."

**8. L366: To me, CDI maps provided in Appendix C are more important than the PCA weighting values (Figure 3). Considering to swap Figures.**

Yes, we agree with your thoughts. In the revised manuscript, we have swapped the figures: Appendix C is now Figure 3, and vice versa. (Due to this change, the earlier appendix sequence has been slightly modified to maintain the flow of the write-up. What was previously Appendix B is now Appendix C, and the former Figure 3 is now Appendix B.

9. L565: Figure 9. For the legend, I think better if you write: >0.5 instead of 0.5<

Thank you for pointing out this. In the revised manuscript, >0.5 has been used in figure 9. The updated figure is as follows:

[Figure]

**Figure 9.** Spatial correlation between CDI and single input-based traditional indices for a sample month (April): (a) CDI vs. SPI, (b) CDI vs. STI, (c) CDI vs. NDVI, and (d) CDI vs. SSMI. Negative correlations are represented in shades from yellow to red, while positive correlations are shown in shades from light green to dark green.

10. L569: Aftereffect or after effect?

Thank you for the suggestion. We have revised the text to use "after effects" as recommended.

11.L584: Maybe provide reference from WMO (2012)? https://library.wmo.int/records/item/39629-standardized-precipitation-index-user-guide.

Thank you for the suggestion. The reference has been given to the sentence as follows:

"SPI-1 and SPI-3 are effective for detecting meteorological and agricultural droughts, respectively, while longer time scales (e.g., SPI-6 or SPI-12) are more suitable for identifying hydrological droughts (World Meteorological Organization, 2012)."